# A composite approach to produce reference datasets for extratropical cyclone tracks: Application to Mediterranean cyclones

Emmanouil Flaounas[1], Leonardo Aragão[2], Lisa Bernini[3], Stavros Dafis[4,5], Benjamin Doiteau[6,7], Helena Flocas[8], Suzanne L. Gray[9], Alexia Karwat[10], John Kouroutzoglou[8,11], Piero Lionello[12], Mario Marcello Miglietta[13], Florian Pantillon[6], Claudia Pasquero[3], Platon Patlakas[8], Maria Angeles Picornell[14], Federico Porcù[2], Matthew D. K. Priestley[15], Marco Reale[16,17], Malcolm J. Roberts[18], Hadas Saaroni[19], Dor Sandler[19], Enrico Scoccimarro[20], Michael Sprenger[21], Baruch Ziv[19,22]

1 Institute of Oceanography, Hellenic Centre for Marine Research, Athens, Greece
2 Department of Physics and Astronomy "Augusto Righi", University of Bologna, Bologna, Italy
3 Università di Milano - Bicocca, Milano, Italy
4 National Observatory of Athens, Institute of Environmental Research and Sustainable Development, Athens, Greece
5 Data4Risk, Paris, France
6 Laboratoire d'Aérologie, Université de Toulouse, CNRS, UPS, Toulouse, France
7 CNRM, Météo-France and CNRS, Toulouse, France
8 Department of Physics, National and Kapodistrian University of Athens, Athens, Greece
9 University of Reading, Reading, UK.
10 Meteorological Institute, University of Hamburg, Hamburg, Germany
11 Hellenic National Meteorological Service, Hellinikon, Greece
12 University of Salento, Lecce, Italy
13 Institute of Atmospheric Sciences and Climate (CNR-ISAC), National Research Council of Italy, Padua, Italy
14 Agencia Estatal de Meteorología, AEMET, Palma, Spain
15 Department of Mathematics and Statistics, University of Exeter, Exeter, UK
16 National Institute of Oceanography and Applied Geophysics-OGS, Trieste, Italy
17 Abdus Salam International Centre for Theoretical Physics, ICTP, ESP Group, Trieste, Italy
18 Met Office, Exeter, UK
19 Porter School of the Environment and Earth Sciences, Tel Aviv University, Tel Aviv, Israel
20 Fondazione Centro Euro-Mediterraneo sui Cambiamenti Climatici - CMCC, Bologna, Italy
21 Institute for Atmospheric and Climate Science, ETH Zurich, Zurich, Switzerland
22 Department of Life and Natural Sciences, The Open University of Israel, Raanana, Israel

## Abstract

Many cyclone detection and tracking methods (CDTMs) have been developed in the past to study the climatology of extratropical cyclones. However, all CDTMs have different approaches in defining and tracking cyclone centers. This naturally leads to cyclone track climatologies of inconsistent physical characteristics. More than that, it is typical for CDTMs to produce a non-negligible amount of bogus tracks which can be perceived as "false positives", or more generally as CDTM artifacts, i.e. tracks of weak atmospheric features that do not correspond to large or mesoscale vortices. Lack of consensus in CDTM outputs and the inclusion of significant amounts of bogus tracks therein, has long prohibited the production of a commonly accepted reference dataset of extratropical cyclone tracks. Such a dataset could allow comparable results on the analysis of storm track climatologies and could also contribute to the evaluation and improvement of CDTMs.

To cover this gap, we present a new methodological approach that combines overlapping tracks from different CDTMs and produces composite tracks that concentrate the agreement of more than one CDTM. In this study we apply this methodology to the outputs of 10 well-established CDTMs which were originally applied to ERA5 reanalysis in the 42-year period of 1979-2020. We tested the sensitivity of our results to the spatio-temporal criteria that identify overlapping cyclone tracks, and for benchmarking reasons, we produced five reference datasets of subjectively tracked cyclones.

Results show that climatological numbers of composite tracks are substantially lower than the ones of individual CDTM, while benchmarking scores remain high (i.e. counting the number of subjectively tracked cyclones captured by the composite tracks). This suggests that our method is able to filter out a large portion of bogus tracks. Indeed, our results show that composite tracks tend to describe more intense and longer-lasting cyclones with more distinguished early, mature and decay stages than the cyclone tracks produced by individual CDTMs. Ranking the composite tracks according to their confidence level (defined by the number of contributing CDTMs), it is shown that the higher the confidence level, the more intense and long-lasting cyclones are produced. Given the advantage of our methodology in producing cyclone tracks with physically meaningful, distinctive life stages and including a minimum number of bogus tracks, we propose composite tracks as reference datasets for climatological research in the Mediterranean. The supplementary material provides the composite Mediterranean tracks for all confidence levels and in the conclusion we discuss their adequate use for scientific research and applications.

## 1. Introduction

A weather feature may refer to any meteorological system that can be distinguished from its environment using a single, or a combination of atmospheric variables. Such features span scales from local convective cells to planetary waves and may relate to the instantaneous state of the atmosphere or its temporal evolution. Cyclones, both tropical and extratropical, are plausibly the weather features that attract the most scientific attention. The systematic identification and tracking of cyclone centers is indeed a procedure of high interest for issuing warnings of imminent high-impact weather, but also for understanding future tendencies of extreme events and other climate processes (e.g. Ulbrich et al., 2009, 2013; Zappa et al., 2013; Reale et al., 2022).

Over the past few decades, several methods have been developed to systematically detect and track cyclone centers in gridded datasets. Cyclone detection and tracking methods (hereafter CDTMs) are based on a series of arbitrary choices about (i) the atmospheric variables that best describe cyclones, (ii) the preprocessing operations applied to their fields, (iii) the criteria that define cyclone centers and (iv) the adopted approaches to track cyclone centers in time. Despite their differences, all CDTMs follow a two-step procedure: first, all methods need to define the representative location of cyclone centers and, second, tracks need to be built by connecting the identified cyclone centers in consecutive time steps. Methodological approaches in both of these steps are crucial for the quality of the produced cyclone tracks.

In the first step, cyclone centers are typically defined as local maxima of relative vorticity, or as local minima of geopotential height or mean sea-level pressure (MSLP). However, locations of cyclone centers may differ significantly among CDTMs, even if the same input fields are used (Sinclair, 1994; Neu et al., 2013). This is due to the application of additional criteria (e.g. application of threshold values and spatial gradient fields) or the use of spatial and temporal filters that smooth the fields or remove tracks over high orographic features (Hoskins and Hodges, 2002; Hanley and Caballero, 2012; Neu et al., 2013; Messmer et al., 2015). The definition of cyclone centers is of paramount importance for the physical characteristics of the produced tracks. If strict criteria are applied to the input fields or strong spatial filters are used to remove noise therein, only cyclone centers of deep MSLP or high vorticity will be identified. As a result, tracks will most plausibly include well-organized cyclone systems, but other important shallower systems will be omitted. In addition, all produced tracks will tend to be limited to times close to cyclones' mature stage (i.e. time of maximum intensity defined by the track point of lowest MSLP, or highest maximum vorticity) since cyclone centers in early and late stages will be discarded or filtered out by the method's strict criteria and preprocessing procedures. On the other hand, less strict criteria produce a large number of "bogus tracks", which can be perceived as "false positives", or more generally as CDTM artifacts. Bogus tracks might correspond to persistent weak MSLP perturbations or long-lasting vorticity local maxima

due to abrupt wind steering (e.g. close to steep topographic features). More generally they can hardly be interpreted as well-organized vortices.

In the second step, all CDTMs connect centers that have been found in successive time-steps to describe the displacement of the same single cyclone system. In this procedure, the CDTMs usually adopt a translation speed limit, i.e. the maximum distance between two cyclone centers in consecutive time steps. This criterion is strongly dependent on the time interval of the input fields (Crawford et al., 2021; Aragão and Porcù, 2022): short time intervals between input fields (e.g. hourly fields) require smaller translation limits and vice versa. If more than one cyclone center is located within this limit, the CDTMs have to choose which corresponds to the track's natural continuation. The more cyclone centers are identified in the first step of the CDTMs (e.g. due to less strict definitions of cyclone centers), the higher the probability that the methods choose the "wrong" cyclone centers to connect in the second step. Setting a small translation limit diminishes the number of candidate cyclone centers that could continue the tracks, but it is then more likely that the CDTM will fail to capture the full extent of tracks of fast-moving systems. In these regards, the spatial resolution of input fields is also a crucial factor for the quality of the produced tracks (Kouroutzoglou et al., 2011). For instance, using high spatial resolution might lead to several local minima of MSLP being nested within a single large-scale cyclonic system. All these local minima might be identified as "distinct cyclone centers". In such cases, CDTMs either produce abrupt "jumps"of track points or describe the displacement of single cyclone systems with more than one track.

The IMILAST project (Neu et al., 2013) has performed a comprehensive intercomparison of CDTMs showing disagreement and consensus among methods and discussing weaknesses and advantages that depend on the nature of the tracked cyclone. In fact, cyclone climatologies produced by individual methods often differ significantly in the number of cyclones, track densities, cyclone intensities and temporal trends. When combining track datasets, most methods agree to a great extent on basic features of cyclone climatology and when tracking strong well-organized systems like tropical cyclones (Neu et al., 2013; Bourdin et al., 2022). In other cases, however, methods may capture different parts of the same tracks. In addition, several tracks might be completely missed by individual methods, while a large amount of bogus tracks might be produced.

The natural question that arises from the above is whether different CDTMs might be combined to build datasets of "high confidence". Such datasets would be expected to include: i) composite tracks that were commonly captured (partly or entirely) by individual methods, and ii) the lowest possible number of bogus tracks.

In this study we use a new approach to produce high confidence datasets for the Mediterranean region based on the recent ERA5 reanalysis (Hersbach et al., 2020). Cyclogenesis is frequent in the Mediterranean, producing a high number of shallow and deep cyclones per year (Trigo et al., 1999; Campins et al., 2011; Lionello et al., 2016; Flaounas et al., 2022). However, Mediterranean cyclones are challenging weather features to track, mainly due to their small size when compared to other extratropical cyclones, but also due to the complex geography with sharp land-sea transitions and high mountain chains that surround the Mediterranean basin (Lionello et al., 2016; Flaounas et al., 2018). In fact, lee cyclogenesis is frequent and cyclone systems often cross continental areas, distorting their MSLP and relative vorticity structures (Buzzi and Tibaldi, 1978; Buzzi et al., 2020). As a result, atmospheric variables that typically describe cyclones present high spatial variability that challenges the CDTM performance, especially in high-resolution datasets (Ruti et al., 2016).

The following section presents our methodological approach and the procedure for benchmarking the tracks. Then, we present the physical characteristics of cyclone tracks produced by individual CDTM and compare them to the ones of composite tracks. Finally, we discuss the advantages in using composite tracks as reference datasets, compared to tracks from individual CDTMs. This paper ends with dataset availability and conclusions on the use of composite tracks of different confidence levels for scientific research.

## 2. Datasets and methods

### 2.1 Building composite tracks: The methodological approach through two cyclone cases

In this study, we use 10 CDTMs (further referred to as M01 to M10), briefly described in the Appendix and summarized in Table 1. All 10 CDTMs were applied to hourly ERA5 reanalysis fields with a regular grid spacing of 0.25°x0.25° in longitude and latitude. In contrast to other reanalysis, ERA5 is available in fine grid spacing allowing CDTMs to track small scale cyclones. Furthermore, the availability of hourly fields is advantageous for the process of tracking.

Each CDTM produced cyclone tracks for the 42-year period of 1979-2020 within a rectangular domain encompassing the broader Mediterranean region, defined by 20°N-50°N and 20°W-45°E. All tracks have been produced in 13-month intervals starting from the 1 January of a given year and ending on 31 January of the following year. This was done to avoid track discontinuities on 31 December. Following the IMILAST protocol (Neu et al., 2013) tracks that lasted less than a day (with less than 25 trackpoints) have been discarded to exclude short-lived cyclonic features. Moreover, for the sake of homogeneity in measuring a cyclone's intensity, the MSLP was extracted as the minimum MSLP within a radius of 2.5 degrees from the same track points obtained by each CDTMs, regardless of their input field in Table 1.

All tracks from the 10 CDTMs have been used to build the composite tracks in a three-step procedure that is summarized in the flowchart shown in Fig. 1.

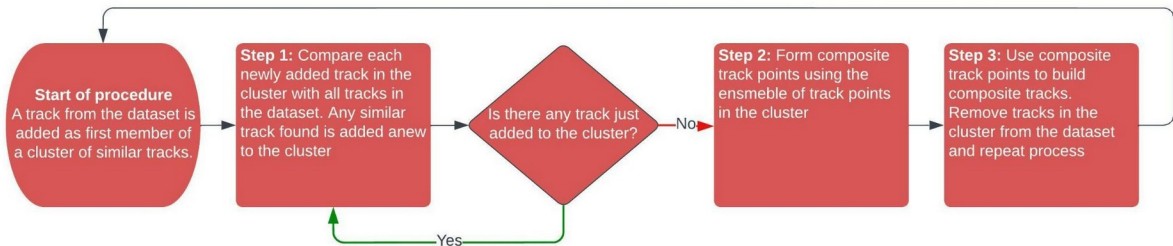

**Figure 1** Flowchart of the procedure that builds composite tracks

*1st step:* The algorithm starts from the first track and searches for all other "similar" tracks within the dataset containing all the tracks from all the CDTMs. Two tracks are identified as "similar" if their track points overlap in space and time. Since the overlap of two tracks is unlikely to be perfect, space and time threshold criteria are applied. In terms of space, two track points belong to similar tracks if they occur at the same time and are no more than 300 km apart. This distance threshold was chosen as a length order of the minimum radius of Mediterranean cyclones (Campins et al., 2011; Reale et al., 2022). Identified cyclone centers further apart than 300 km could belong to either distinct cyclones, or distinct centers, nested within relatively large cyclonic circulations. Sensitivity tests have been performed for a distance threshold of 500 km with minimum impact on the final results. The temporal overlapping criterion refers to the number of grid points that belong to the overlap between two tracks (i.e. the time period in which two tracks share the same segments). To test the sensitivity of results to the temporal criterion, in this study we qualify two tracks as similar if they overlap by 6, 12, 18, or 24 hours (i.e. if they share 7, 13, 19 or 25 grid points).

Figures 2a and 2b show the outcome after applying step 1 of our method to two cyclone cases. For these two cases, we choose a temporal overlapping threshold of 24 hours. Therefore, every track shown in Figs 2a and 2b overlaps with at least another track by at least a day, i.e. shares at least 25 similar track points. Clearly in Fig. 2a, all cyclone tracking methods have captured similar tracks. On the other hand, several individual tracks in the second cyclone case (Fig. 2b) have different lengths within the illustrated domain. It is noteworthy that in step 1, each ensemble track may join different tracks of a single CDTM. For example, Fig. 2b includes 15 cyclone tracks in total, where several

methods captured one track in the Western Mediterranean and a second one over the Eastern Mediterranean, that overlap with the same rather long track of M07.

*2nd step*: In this step, composite track points are created at the average locations of all track points identified as similar in step 1 (i.e. track points that share the same time and are not 300 km apart). The number of methods used to create composite track points defines the "confidence level", which ranges
from 2 to 10. For instance, a confidence level of 5 suggests that a track point was captured by at least five CDTMs. As such, final composite track datasets of any confidence level are always subsets of datasets with lower confidence level. The confidence level of the composite track point is depicted by the size of black and red dots in Figs 2c and 2d. Clearly, the middle sections of the tracks tend to concentrate composite track points of higher confidence level with respect to the edges, where fewer
tracks are close to each other. Presumably, when more CDTMs have identified similar track points, the less likely these track points make part of bogus tracks as discussed in Section 1.

*3rd step:* Starting from composite track points with the highest confidence level, we build all possible cyclone tracks by connecting composite track points forward and backward in time. If more than one composite track point is available to continue building a composite track, then our method chooses the
one with highest confidence level or the closest one if confidence levels are equal. Three conditions are necessary to connect two composite track points: (i) they must take place in consecutive time steps; (ii) they have to be located within a threshold distance; and (iii) two consecutive composite track points cannot have confidence level of 1. The threshold distance ranges from a minimum of 300 km (i.e. the threshold distance that identifies similar track points) to a maximum that is defined by the
maximum distance of consecutive track points from all tracks that contribute to the composite track points (provided this maximum distance exceeds 300 km). A minimum of 300 km allows continuation of composite track points that were produced by different CDTMs. A maximum value allows our method to always adapt to the particular configurations of the participating CDTMs. The condition that two consecutive track points cannot have a confidence level of one is applied to avoid
reproducing tracks that were captured from a unique CDTM and consequently could correspond to a bogus track. If step 3 produces more than one composite track, we eventually retain the one that includes track points with the highest average level of confidence.

As an example, Fig. 2e shows that the composite track is similar to most tracks in Fig. 2a. It is noteworthy that outlier track segments such as those from M07 over Northwest Africa, from M08
over the Mediterranean Sea and few unrealistic "jumps" of M01 towards the easternmost part of the tracks (over Egypt) have a limited effect on the final composite track. In contrast to this cyclone case, the CDTMs in Fig. 2b lacks the required consensus for the production of a single dominant cyclone track. In fact, the composite track in Fig. 2f neglects the ensemble of tracks in the eastern Mediterranean. This is due to the third constraint of step 3. In fact, our method started building several
composite tracks from the Western Mediterranean where confidence level is high (black dots in Fig. 2d). Continuity of these composite tracks towards the Eastern Mediterranean (red dots in Fig. 2d) would rely on a single cyclone tracking method (M07) and step 3 prohibits the connection of multiple track points with confidence level of 1. For the same reason, the composite track omitted the ensemble of westernmost tracks, as also the northernmost and easternmost extensions of M08 and the
southernmost part of M04. It is noteworthy that all tracks found to be similar in step 2 are discarded even if they did not contribute to the composite tracks. As a result, the omitted ensemble of tracks in the eastern Mediterranean (Fig. 2f) were not later used to produce a new composite track of lower confidence level.

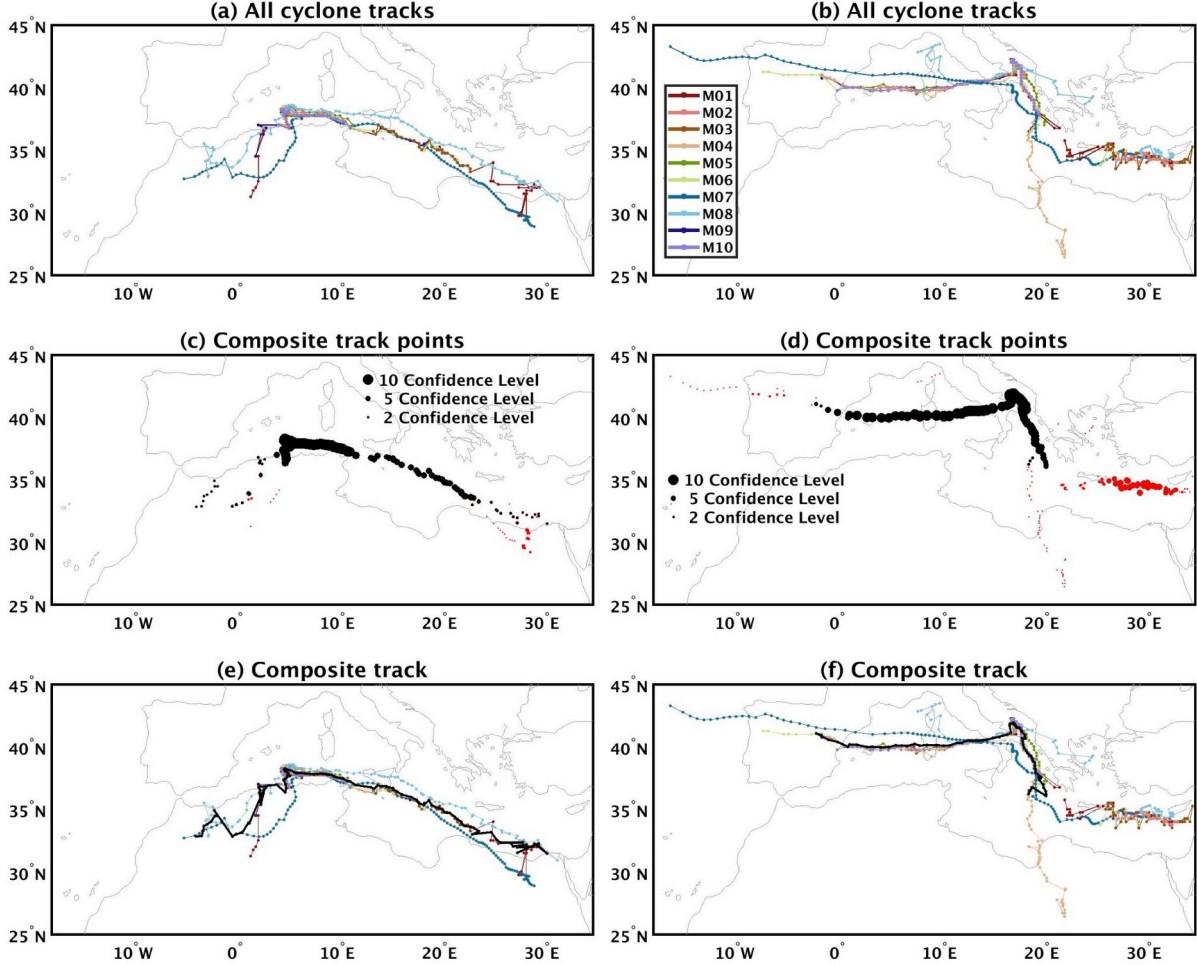

**Figure 2 a** Step 1 of the method (see text): similar cyclone tracks that have been reproduced by 10 different CDTMs for the cyclone case of November 2020. **c** Step 2: Composite track points produced by the combination of the tracks in **a**. Black (red) color marks the composite track points that have (not) been eventually used for composite tracks. The size of the dots depicts the confidence level of the composite track points. **e** Step 3: as in **a**, but overlaying the composite track (in black line). **b**, **d**, and **f** as in **a**, **c**, and **e** respectively but for a second cyclone case that took place in November 2018.

After building a composite track, all of its composite track points have been assigned to the lower MSLP value within a 2.5 degrees circular area. This operation is adequate to identify different stages of a cyclone's lifecycle (e.g. intensification, mature stage and decay). Figure 3a shows the MSLP evolution of the composite track for the cyclone case in Fig. 2a along with the MSLP evolution of the tracks from the 10 CDTMs. It is rather clear that all tracks reproduce similar dynamical lifecycles suggesting that the composite track can be used for a meaningful analysis of different cyclone stages: from a weak low-pressure system until the decay of deep mature cyclones. For the first cyclone case, Figs 3c and 3e show the MSLP fields for hour times 15 and 35 respectively (vertical lines in Fig. 3a) along with parts of the tracks from all 10 CDTMs that eventually contributed to the production of the composite track in Fig. 2e. The black dot in Figs 3c and 3e shows that the composite track point (black dot) at the times depicted by vertical lines in Fig. 3a is meaningfully close to a local minimum of MSLP. However, this should not be a surprise since all tracks from individual CDTMs are fairly well consistent with each other (Fig. 2a). In the second case, Fig. 3b shows that the composite track, as in Fig. 3a, captures both the deepening and decay stages of the cyclone. For this second case, we select two different times: one at the end of the lifetime of the composite track (time 94 in Fig. 3b) and another one 12 hours later, at time 106. Figure 3d shows that the composite track point is still consistent with a MSLP local minimum, while another minimum is followed by the tracks in the

eastern Mediterranean. Figure 3f clearly shows that the cyclone tracks in the eastern Mediterranean are still following the same MSLP local minimum while the composite track ceased to exist along with the local MSLP minimum that it was following in Fig 3d.

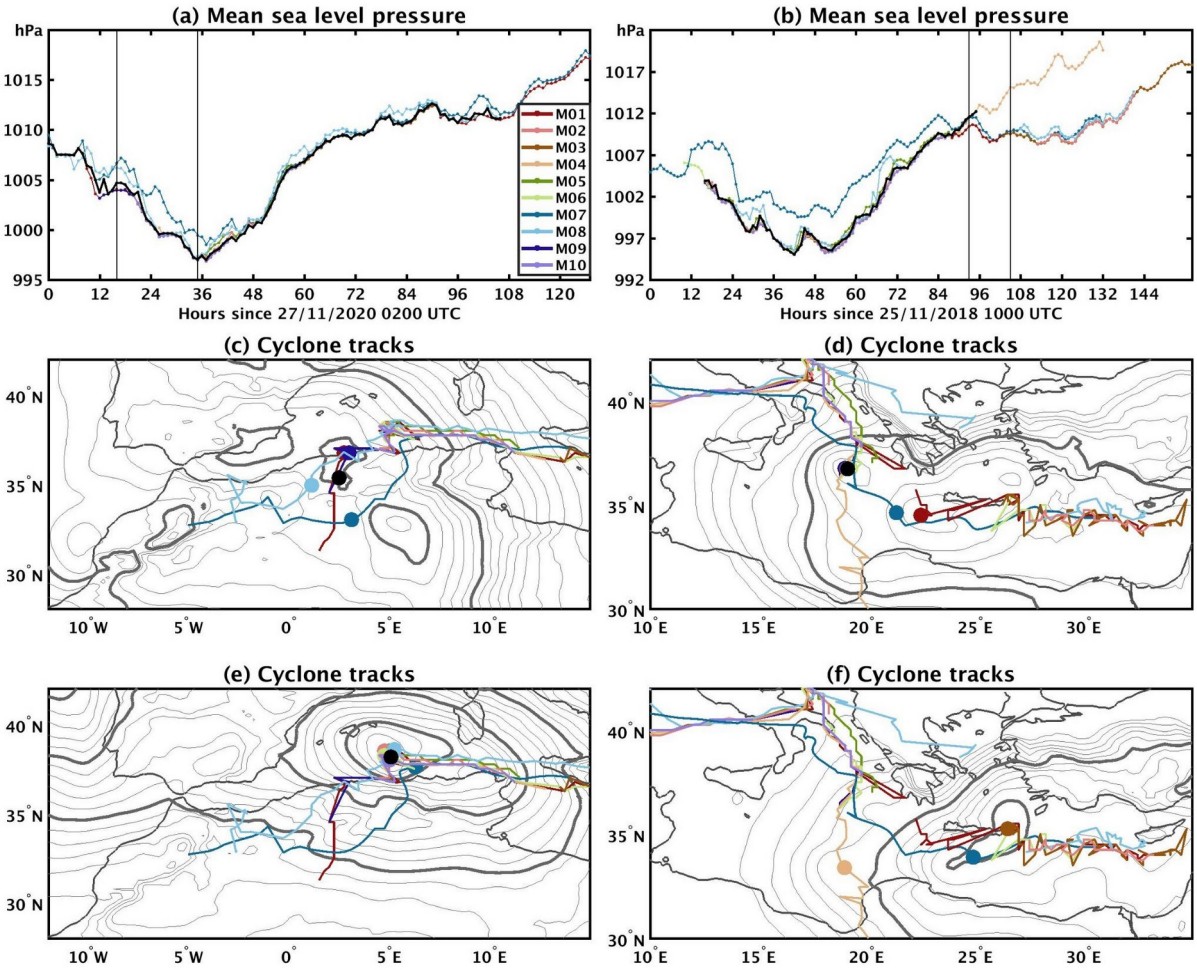

Figure 3 a The MSLP evolution of cyclone tracks for the cyclone case (in coloured lines) of November 2020, shown in Fig. 2a. MSLP of the composite track is overlaid in black color. c MSLP field in gray contours at the time of the first vertical line in panel a (1 hP contour intervals where thick contours are drawn every 5 hPa starting from 1000 hPa). Tracks from different CDTMs are shown in coloured lines. Colored dots depict the location of cyclone centers and the black dot depicts the location of the corresponding composite track point. e As in c but for the time of the second vertical line in panel a. b, d, and f as in a, c, and e but for a second cyclone case that took place in November 2018 shown in Fig. 2b.

## 2.2 Benchmarking the performance of cyclone tracking methods

A major challenge in the field of cyclone tracking is the absence of reference datasets for benchmarking the performance of CDTMs. While best-track datasets are usually issued from forecast services of tropical cyclones, there are no similar datasets for extratropical cyclones and specifically for Mediterranean cyclones. The lack of such best-track datasets can be arguably traced back to the highly variable spatial structure of extratropical cyclones (Neu et al., 2013). Indeed, extratropical cyclones may lack the clearly distinguishable centers which are typically found in their tropical counterparts. For instance, in cases of secondary cyclogenesis or cyclone families, a CDTM might detect two cyclones as one storm. Alternatively a CDTM might only partially identify fast-moving storms and explosive cyclones due to the application of strict constraints in its tracking and detection

procedures. Therefore, the evaluation of CDTMs in climatological studies remains intuitive and largely relies on qualitative evaluation.

In our methodological approach, we use the confidence level of composite tracks as a measure of robustness, i.e. whether composite tracks concentrate high or low agreement among CDTMs. However, it is still an open question whether a low confidence level might lead to the inclusion of a
large number of bogus tracks, or if a high confidence level might exclude well-organized cyclone systems that were captured by few individual CDTMs. As a result, benchmarking performance of the 10 CDTMs and their consequent composite tracks remains a key issue for this study.For this reason, we performed a subjective tracking procedure where five meteorologists (see author contributions) performed subjective manual tracking of 117 selected Mediterranean cyclones, all derived from cases
of the past 40 years. Appendix B provides a short description of the origin and track of the 117 cyclones, while supplementary material 1 provides a list of their dates and scientific references where relevant.

The subjective cyclone tracking procedure was based on a computer routine that displays hourly MSLP fields from ERA5 and allows the user to manually pinpoint the hourly position of each cyclone
center using their own subjective criteria. A more complete approach would also require the use of relative vorticity fields. However, this option was not selected due to the spatial noise of this field and the requirement of additional post-processing procedures to easily distinguish cyclone centers. The five meteorologists were instructed: (i) to only document the clearest possible cyclone center displacements, (ii) to stop the tracking if the cyclone centers performed unreasonable displacements,
i.e. spatial "jumps", (iii) to only retain tracks that lasted at least 24 hours, and (iv) to make sure that all tracks had consecutive hourly track points. Subjective cyclone tracking is, by definition, subject to human errors and may produce different tracks for the same cyclone systems. Moreover, the final datasets are not reproducible. Nevertheless, it is the subjective criteria that transform these tracked cyclones into "useful" datasets, i.e. the included tracks would be potentially selected by a researcher
for reasons of scientific research and for operational forecasting purposes when high impact weather is imminent. It is noteworthy that these tracks have been produced using MSLP fields alone. Therefore, cyclones that would be identified in a clearer way by relative vorticity or other atmospheric variables may be absent from these datasets. In fact, the subjective production of a reference dataset for the ends of CDTMs assessment would demand thorough and rigorous investigation of weather
maps and the use of more atmospheric variables (wind, geopotential height,..). In our more simplistic approach the level of agreement between the subjective tracks is strongly dependent on the different human perceptions of cyclone centers in each expert meteorologist, given the same atmospheric fields. This was done to gain insights into the maximum of agreement that we should also expect from the individual CDTMs, i.e. how much agreement with the subjectively tracked cyclones would be
considered as acceptable. Taking into account the above, the five datasets should be regarded as a reference for benchmarking the ability of individual CDTMs and composite tracks to capture a certain number of well-known cyclone cases. This number is statistically small compared to the number of cyclones in a 42-year climatology and consequently benchmarking can not be considered as a token of the general quality of CDTMs.

The procedure of subjective cyclone tracking produced five datasets composed of 68, 59, 73, 82 and 97 cyclone tracks. It is noteworthy that many cases in the list of 117 cyclones were not clearly distinguished in MSLP fields of ERA5 or were not tracked for at least 24 hours. To quantify the degree of similarity between subjectively tracked cyclones, we apply step 1 of our methodological approach to every pair of datasets: two tracks are defined as similar if they share common track points
for a certain threshold time period. Similarity score is then defined as the number of similar tracks, divided by the smaller number of tracks included in either of the two datasets. A similarity score of 100% implies that the ensemble of tracks included in the smaller dataset may be considered as a subset of the other dataset. Figure 4 shows the similarity scores between the five datasets of subjectively tracked cyclones (hereafter D01 to D05) for four overlapping criteria, i.e. two tracks are
qualified as similar if they overlap by 6, 12, 18 or 24 hours. If we use an overlapping criterion of 6 hours, similarity ranges between 63% and 88% with D01 presenting the least agreement with the

other datasets, and datasets D02 and D04 being the most similar ones. On the other hand, if a 24 hours criterion is used, then similarity ranges between 54% and 81%. The decreasing percentages as a function of the overlapping criterion suggests that the complexity of cyclone systems is evolving in time and therefore meaningful cyclone tracking is also dependent on the stage of a cyclone. Such scores suggest that subjective criteria lead to different perceptions of how a cyclone center might be displaced in time and highlights the necessity for using robustly identified tracks. The scores in Fig. 4 may be used as indicative of the expected level of agreement when performing subjective, manually done, case-to-case analysis. Therefore, two different CDTMs that present a similarity score of the order of 80% with an overlapping criterion of 6 to 18 hours may be roughly considered to have reached the level of similarity that would have been reached also by experts performing subjective analysis.

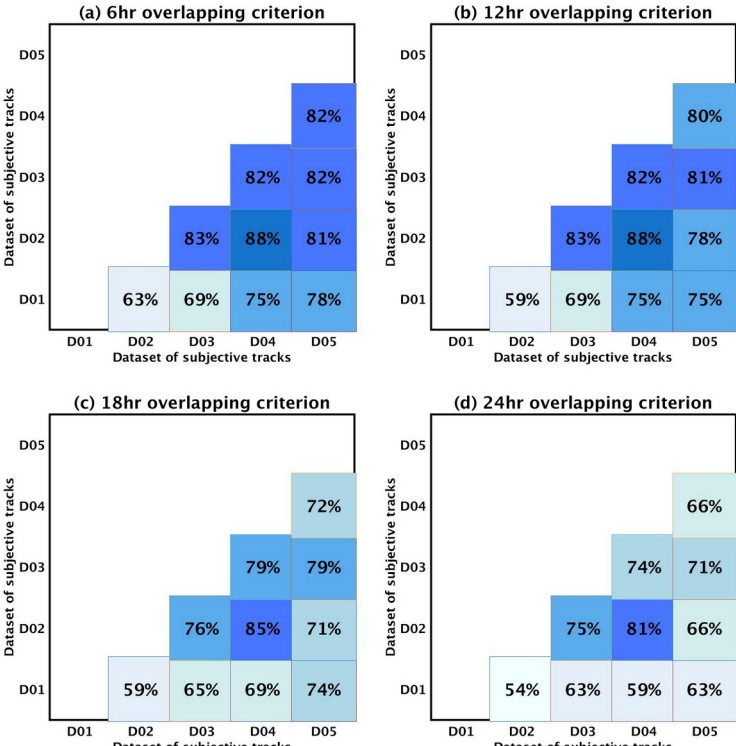

**Figure 4** Similarly scores among the five datasets of subjectively tracked cyclones (D01, D02.., D05), defined as the number of similar tracks in two datasets divided by the number of tracks in the smallest dataset. Each of the four panels presents similarity scores for different overlapping criteria.

## 3. Mediterranean tracks from the perspective of individual CDTM

### 3.1 Physical characteristics of Mediterranean cyclone tracks

The climatology of Mediterranean cyclones has been the object of several studies in the past and such analysis is out of our scope. However, to gain deeper insights into the usefulness of composite tracks, this section presents the diversity of the physical characteristics of Mediterranean cyclone tracks, as produced by the 10 individual CDTMs. This diversity may be attributed to the different configurations of CDTMs, which make them adequate to track cyclone systems of different physical characteristics, but also to CDTMs' sensitivity in producing a statistically important number of bogus tracks. In the following, we present the following track diagnostics: (i) number of tracks per year and season, (ii) spatial density, and (iii) statistical distribution of lifetime, displacement speed and intensity.

- Interannual and seasonal cycle: Figure 5a shows that the number of tracks differs significantly among the methods, with M03 having the largest number of detected cyclones (about 500 per year).

On the other hand, methods M04, M08 and M10 show the lowest number of tracks (approximately 100 to 120 per year). Moreover, there is no clear interannual trend for either method while only a few anomalous years are observed. Figure 5b shows that the seasonal cycle is fairly weak for all the CDTMs. Cyclogenesis tends to be evenly distributed along the year with 7 to 12% per month.
Methods M04 and M09 show prominent seasonal cycles with a minimum (maximum) in (summer) spring. On the other hand, several methods (M01, M03, M06 and M10) show a peak of cyclogenesis during the summer months of the year. Despite these differences, when considering only the 500 most intense cyclones in each method, seasonal cycles of cyclogenesis become more similar. Indeed, Fig. 5c shows a prominent seasonal cycle that comes in agreement with recent studies on Mediterranean cyclones climatology where most intense cyclones tend to occur in winter and spring (e.g Campins et
al., 2011; Flaounas et al., 2015; Lionello et al., 2016). The comparison of Figs 5b and 5c suggests that it is rather "easier" for CTDMs to agree on the seasonal cycle of well-developed intense cyclones than on that of shallow cyclones that are expected to be less distinguishable from bogus tracks making their identification more uncertain. For instance, let a simplistic approach where cyclone centers are defined as grid points of the lowest MSLP among their eight neighboring ones. If the identified
centers correspond to very low values (e.g. 980 hPa), then it is plausible to assume that these centers are related to rather deep cyclones. On the other hand, if centers are close to the regional average (e.g. 1015 hPa), then they likely correspond to rather weak or shallow systems, or simply to local minima of MSLP that are not related to well formed cyclonic circulation, i.e. to a bogus track.

- *Cyclone tracks' density:* Figure 6 shows different patterns of cyclone track densities, calculated at
every grid point as the average count of track points per year within a circular area of radius of 0.5°. It is noteworthy that several CDTMs apply spatial filters to smooth input fields (Appendix A). Therefore, no cyclones are identified at the edges of several panels in Fig. 6 and blank areas depend on the size of the spatial filter. For all CDTMs, higher densities are concentrated over maritime areas, close to the Gulf of Genoa, at the east of the Italian peninsula and over the Adriatic and Ionian Seas.
Other areas of high densities include northwest Africa, the areas close to the Atlas mountain, the Turkish coasts and the eastern side of the Black Sea. All of these cyclogenesis areas have been indeed identified by past studies (Lionello et al., 2016; Flaounas et al., 2018; Reale et al., 2022, Aragão and Porcù, 2022). In these regards, no CDTM produces fully unrealistic results. It is however noteworthy that track densities differ in numbers and for all CDTMs there are locally peaking values that exceed
10 cyclones per year. In fact, M07 and M08 present the smoothest fields, with M07 producing the largest and more distinct centers of high track densities (deep purple colors in Fig. 6). This is usually the case when CDTMs perceive persistent local minima of MSLP and geopotential (or maxima of relative vorticity) as stationary cyclone tracks.

- *Physical characteristics of tracks:* The lifetime of cyclone tracks (Fig. 7a) varies depending on the
CDTMs with 75% of cyclone tracks typically lasting within a range of 24 to 48 hours (i.e. the whole boxplots, excluding whiskers). On the other hand, extreme cyclone duration might reach 72 hours (whiskers). As an exception, M04 and M07 present longer lifetimes with medians and extremes exceeding the duration of 48 hours and 120 hours, respectively. Regarding the average displacement speed of cyclones (i.e. the average distance between hourly consecutive track points per track), Fig.
7b shows that most CDTMs concentrate their average cyclone speed distributions within a range of 0 to 60 km h$^{-1}$. Faster cyclone speeds are found in M01, M04 and M08 probably due to these CDTMs allowing the largest distances between consecutive track points. Finally, in terms of intensity (Fig. 7c), most distributions are concentrated within greater values than 1000 hPa. This result comes in fair agreement with previous studies on Mediterranean cyclone climatologies (e.g. Lionello et al., 2016;
Flaounas et al., 2018; Reale et al., 2021). By design, all CDTMs capture lower MSLP distributions during the cyclone mature stage (middle boxplot in Fig. 7c), defined as the track point of minimum MSLP. However, the overlap of the mature stage distributions of MSLP with those of the initial and decay stages is fairly large. In fact, it is only M04 that exhibits a clear distinction of the three distributions. Such large overlapping suggests that a high number of tracks have no distinct dynamical
lifecycle, plausibly corresponding to bogus tracks, or to partial capturing of tracks (i.e. CDTMs miss large parts of the cyclone dynamical lifecycles).

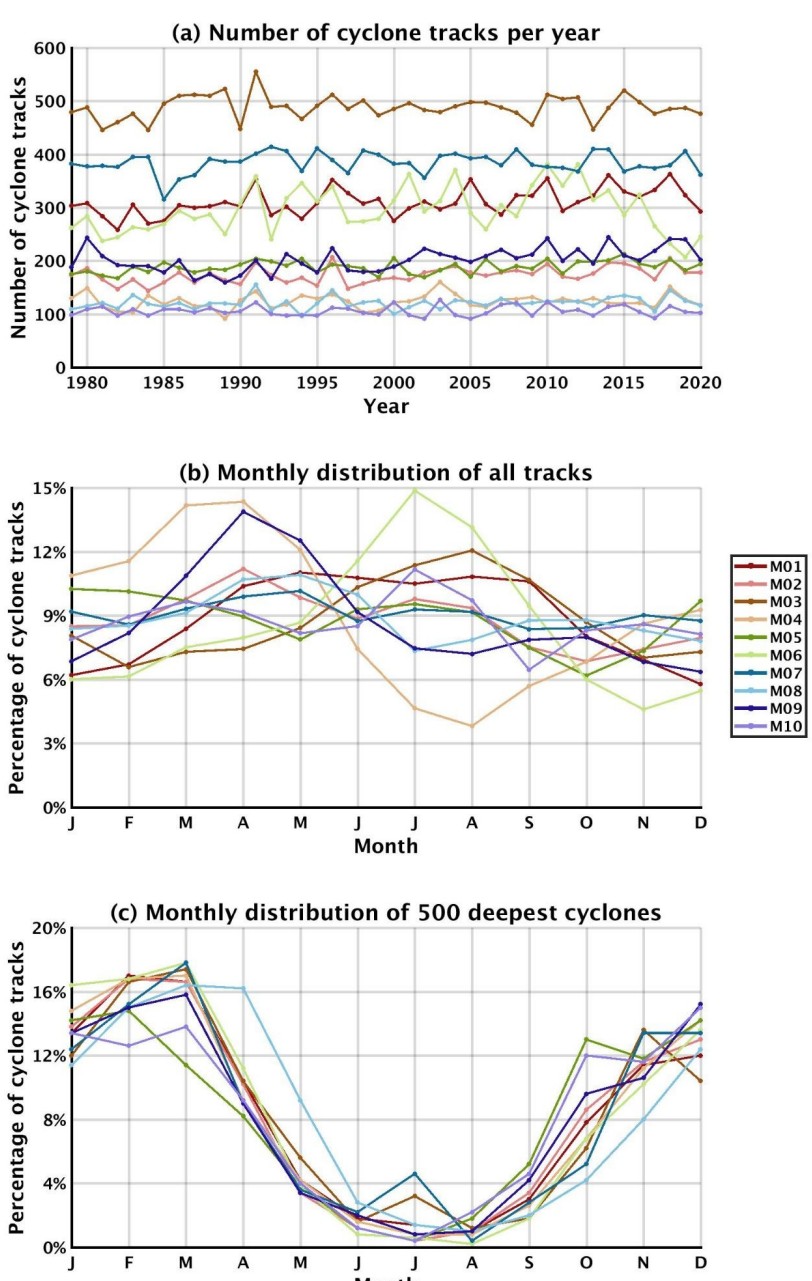

**Figure 5 a** Number of cyclone tracks per year for each CDTM **b** Average monthly distribution of cyclone tracks occurrence for each CDTM with respect to the total number of cyclone tracks per year **c** Monthly distribution of 500 deepest cyclones for each CDTM.


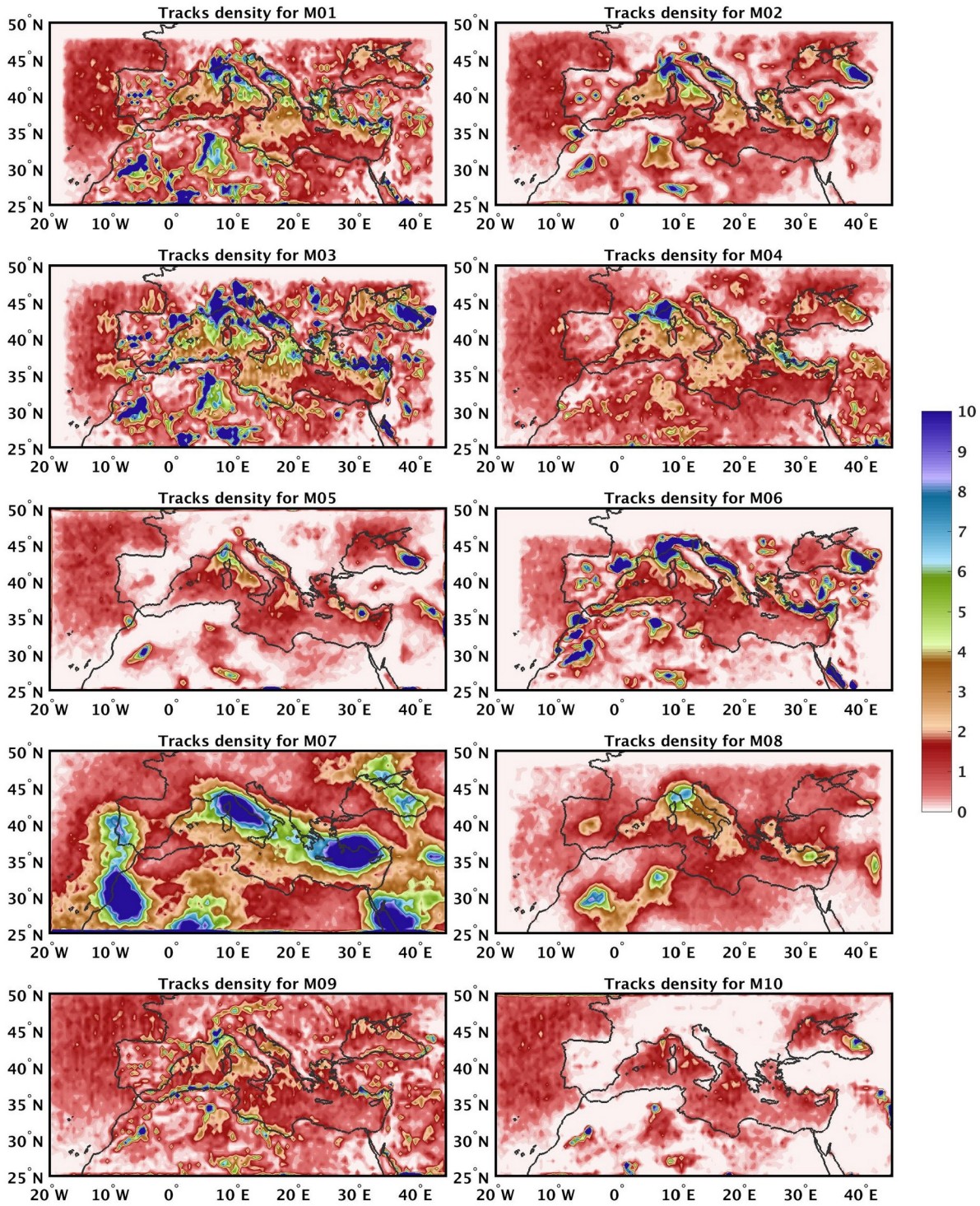

**Figure 6** Spatial density of cyclone tracks for each CDTM. Spatial density expresses the average count of track points per year within a circular area with a radius of 0.5°.

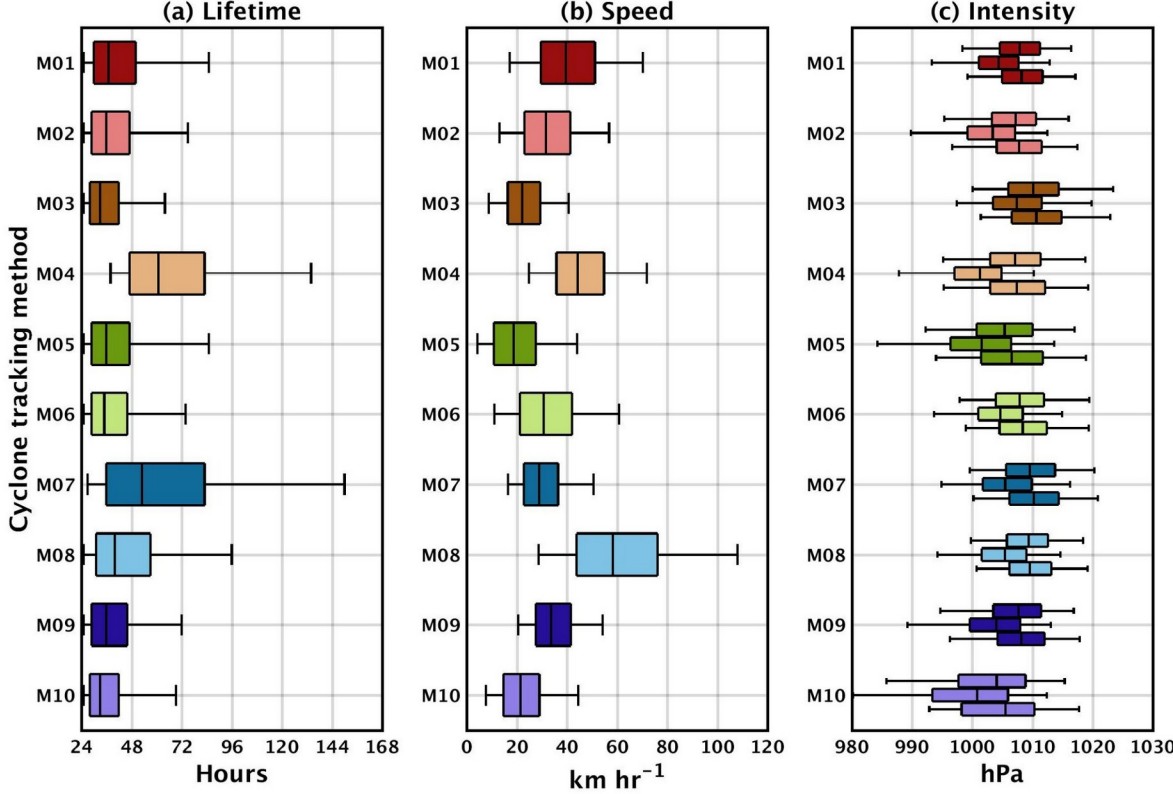

**Figure 7 a** Boxplots showing the distribution of lifetime of cyclone tracks for each CDTM. **b** as in a but for the distribution of average displacement speed per cyclone track. **c** as in **a** but for the distribution of cyclone intensities measured by MSLP. Panel **c** shows three boxplots for every method: the lower one shows the distribution of MSLP at cyclone first track points, the middle one shows the distribution of MSLP at the time of lowest MSLP and the upper one shows the distribution of MSLP at cyclone last track points. For all boxplots the boxes depict the 25th, 50th and 75th percentiles, while whiskers depict the 5th and 95th percentiles.

### 3.2 Similarity of outputs from individual CDTM

As a measure of performance of the 10 CDTMs, Fig. 8 shows the similarity scores between the produced tracks and the ones in the five datasets of subjectively tracked cyclones. Using an overlapping threshold of 24 hours, all methods have consistently tracked more than 65% of the subjectively tracked cyclones. The same percentage exceeds 75% when considering a rather small overlapping threshold of 6 hours. As expected, it is more likely for the CDTMs to capture smaller segments of subjectively tracked cyclones. The spread of scores in Fig. 8 might significantly vary, even by 30% (range of similarity scores is depicted by the spread of whiskers in Fig. 8). Such a high range plausibly reflects the non-negligible disagreement between the subjectively tracked cyclones (Fig. 4) but also the effect of comparing such a small number of tracks to a 42-year climatology. Therefore, the similarity scores in Fig. 8 are not to be taken as a measure of quality for the 10 CDTMs, but as the means to quantify their performance for selected case studies. The results in Fig 8 are thus a reference of individual CDTMs' performance to better understand the quality of composite tracks as reference datasets in the next section. It is noteworthy that several CDTMs were developed and substantially tested for studies in the Mediterranean region (Appendix A). This could result in a more favorable calibration for Mediterranean cyclones. Especially concerning the performance of

M07, this is the only CDTM that uses relative vorticity as an input field while the manual tracks were identified using MSLP. This inconsistency may also have an impact on the results of Fig. 8.

The performance of CDTMs in Fig. 8 suggests that all methods are able to capture most of the subjectively tracked cyclones. However, Fig. 9 shows that similarity between CDTMs is producing
significantly lower scores. As in Fig. 8, the scores tend to decrease while the overlapping criterion is increasing. The highest scores in Fig. 9 are found between the pairs of CDTMs M01-M02 and M01-M08. The pair M01-M02 demonstrates distinctively high similarity, reaching to a score of 67% even when considering overlapping criteria of 24 hours (i.e. two thirds of tracks in M02 overlap with tracks of M01 for at least 24 hours). The lowest similarity scores are found for an overlapping criterion of 24
hours, when comparing M07 with other CDTMs (scores are less than 20%). Being the method that produces longer tracks (Fig. 7a) and largest coverage of the domain with high track densities (Fig. 6), it would be expected for M07 to have higher similarities with other CDTMs. However, M07 is the only CDTM that uses solely relative vorticity to define cyclone centers. It is thus plausible that identifying similar cyclone centers between M07 and the other methods is less favored. With few
exceptions, similarity scores in Fig. 9 rarely exceed 60%, even for a modest overlapping criterion of 6 hours. Given the large similarity scores in Fig. 8, it is plausible to suggest that all CDTMs are adequate to capture intense, well-organized cyclone systems. Therefore, lack of high similarity scores in Fig. 9 could be attributed to the production of a non-negligible number of bogus tracks by each CDTM, or at least tracks that are only captured due to the unique configuration of each CDTM.

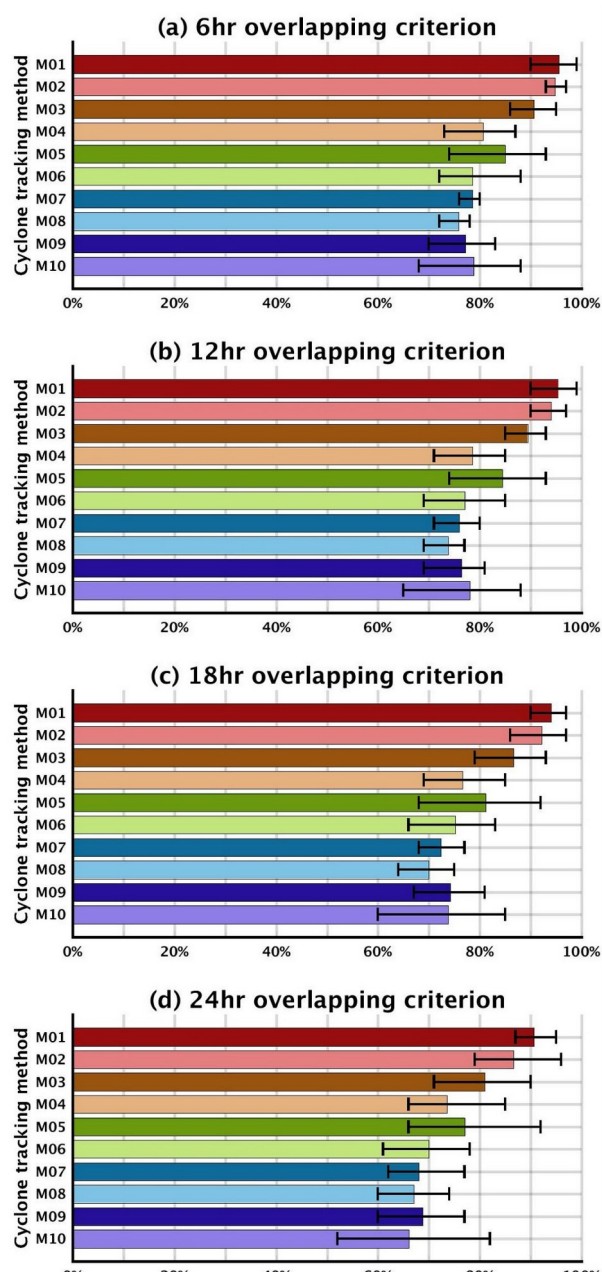

**Figure 8** Bars depict the average similarity score when comparing each of the 10 CDTMs to the five datasets of subjectively tracked cyclones. Minimum and maximum scores are depicted by whiskers in black colors. Each of the four panels presents similarity scores for different overlapping criteria.

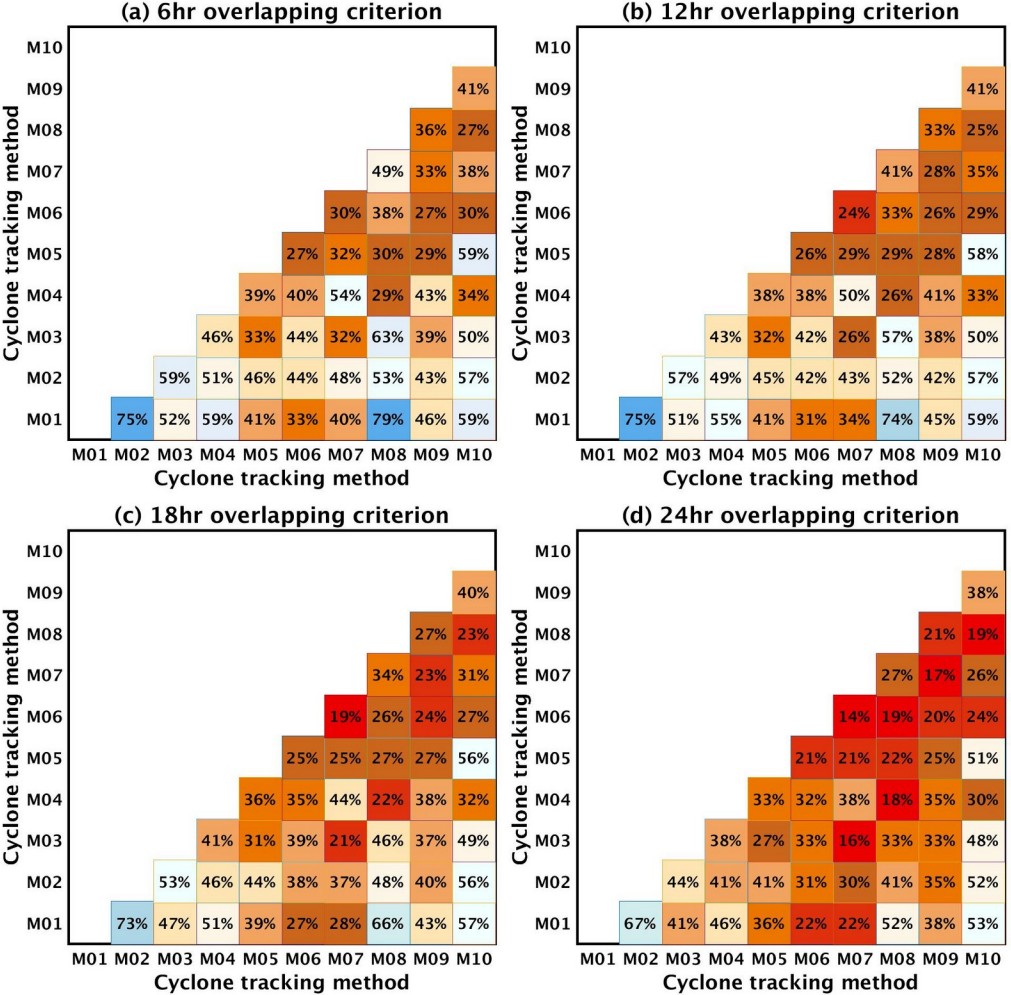

**Figure 9** Similarity scores between the 10 CDTMs, defined as the number of similar tracks in two datasets divided by the number of tracks in the smallest dataset. Each of the four panels presents similarity scores for different overlapping criteria.

## 4. Composite tracks compared to other tracks from individual CDTM

We applied our compositing approach described in Section 2.1 to the ensemble of tracks of the 10 different CDTMs using 4 overlapping criteria and 9 confidence levels (from 2 to 10). This resulted in the production of 36 datasets of composite tracks built by combining at least 2, 3, 4,... or 10 CDTMs that overlap at least 6, 12, 18 or 24 hours. Figure 10 shows the similarity scores between these 36 datasets and the subjectively tracked cyclones. Given our motivation to produce a reference track dataset composed of well-defined and long-lasting intense Mediterranean cyclones, in Fig. 10 we only consider an overlapping threshold of 24 hours, i.e., scores in Fig. 10 show the percentage of subjectively tracked cyclones that overlap with composite tracks in the 36 datasets by at least 24 hours.Clearly the similarity scores and the number of composite cyclone tracks (shown at the bottom of Fig. 10) tend to decrease as a function of the confidence level. This suggests that there is a low probability for a high number of CDTMs to successfully track the same systems. In fact, datasets of higher confidence levels are subsets of datasets with lower confidence levels. For confidence levels up to 7 similarity scores range high, from ~80% to ~90%, compared to an average of 75% in Fig. 8d. It is also noteworthy that similarity scores in Fig. 10 are fairly similar for all four time threshold criteria. It is thus rather plausible to suggest that the overlapping criterion may have a limited effect on the

results of Fig. 10. As an exception, similarity scores for an overlapping time threshold of 24 hours (brown line in Fig. 10) are distinctively lower for confidence levels of 8 by about 8%.

Given the results in Fig. 10, it is rather difficult to define an optimal overlapping criterion or a threshold confidence level that optimizes the detection of all possible well-defined cyclone tracks and rejects all possible bogus tracks. A six-hour overlapping criterion and very low confidence levels may be inadequate for building composite tracks. Indeed, the typical lifetime of intense Mediterranean cyclones is exceeding the order of a day and thus short-time criteria (e.g. 6 hours) would be only adequate if all CDTMs were expected to capture small and different segments of actual cyclone tracks. On the other hand, a 24-hour criterion might be quite close to the characteristic lifetime of cyclones and is thus expected to filter out several important systems in very high confidence levels. Given the similar performance of overlapping criteria in Fig. 10 and the characteristic lifetime of Mediterranean cyclones, we consider an overlapping criterion of 12 hours as an adequate time period for the production of composite tracks. In the following, we present the physical characteristics of composite tracks and we compare them to the ones of individual CDTMs.

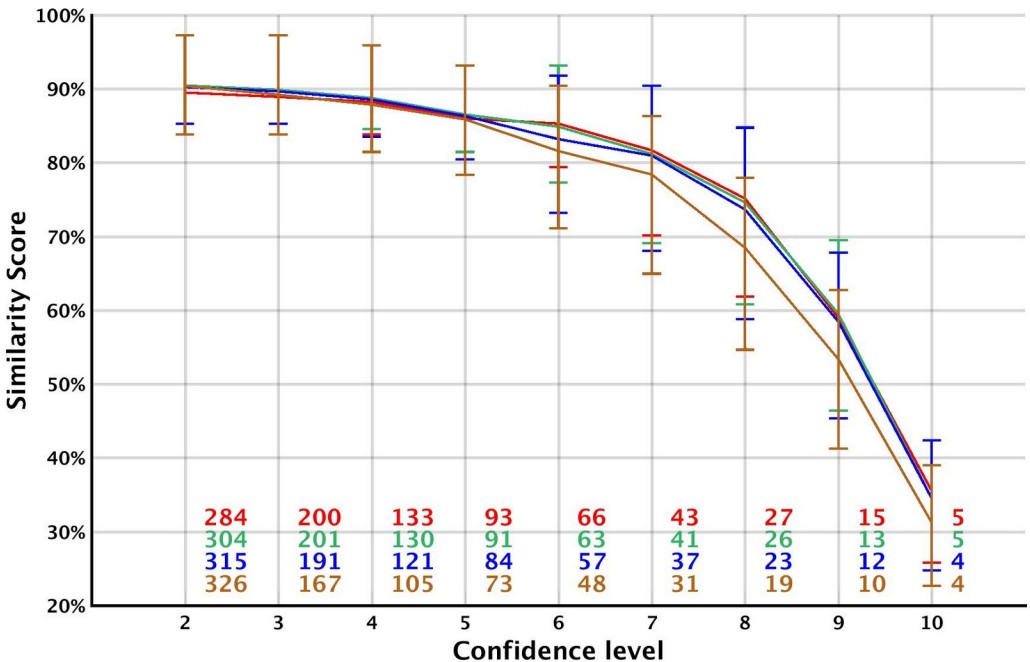

**Figure 10** Similarity scores per confidence level between composite tracks and five datasets of subjectively tracked cyclones. Similarity scores are defined as the number of similar tracks in two datasets divided by the number of subjectively tracked cyclones. Scores are produced for composite tracks, built with four overlapping criteria of 6 (in red), 12 (in green), 18 (in blue) and 24 hours (in brown). Lines show averages and whiskers show minimum and maximum similarity scores. Numbers show the average number of cyclones per year for each confidence level and overlapping criterion.

In order to provide further insights into the contribution of each individual CDTM to the resulting composite tracks, Fig. 11 shows the contribution (in %) of each CDTM to composite tracks (using the 12 hours criterion). For instance, 72% of composite tracks of confidence level 6 were produced with contributions from M06. By design, the composite tracks of confidence level 10 require the participation of all CDTMs. Therefore, contributions of each CDTM to the highest confidence level is

by default 100%. On the other hand, confidence level 2 demands only two CDTMs to agree on the
detection of a specific track. This naturally yields lowest percentages for all CDTMs in Fig. 11 with
M04, M08 and M10 contributing to the composite tracks of confidence level 2 only by 27%. In fact, it
is more likely for a low number of sensitive CDTMs to detect shallow or weak cyclonic systems
whereas another CDTM would fail if it applied more strict criteria. M01 and M03 contribute to more
than half of composite tracks even in datasets with low confidence level. Beyond confidence level 5,
all CDTMs contribute to the majority (>50%) of composite tracks with an average contribution of
85% to composite tracks of confidence level 7. Finally, results in Fig. 11 seem to be independent of
the season although contributions tend to be slightly smaller (higher) in summer (winter) for
confidence levels less than 4 (not shown).

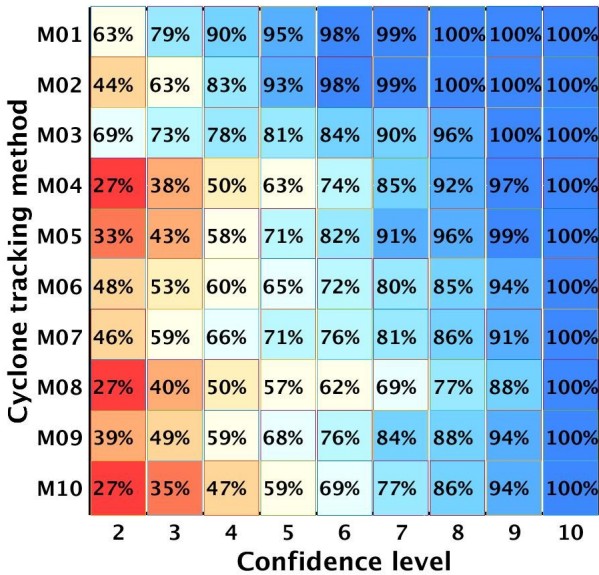


Figure 11 Contribution of CDTMs to the production of composite tracks using the 12 hours criterion.

- *Interannual and seasonal cycles:* Figure 12a shows a gradual decrease of cyclone numbers by about
one third per confidence level. About 300 cyclones per year are found for confidence level 2 and
about 10 cyclones per year when using a confidence level of 10. Annual time series of consecutive
confidence levels in Fig. 12a are significantly correlated with coefficients exceeding 0.8. The highest
correlation coefficient is found between confidence levels 4 and 5, reaching to 0.89, and the lowest for
confidence levels between 8 and 9 with a value of 0.58. The higher the confidence level, the more
cyclones are filtered out from the time series and correlations of annual time series are likely to
become weaker. Nevertheless, Fig. 12a suggests that all time series tend to retain a rather common
interannual distribution of cyclone occurrences. In contrast, Fig. 12b shows that the seasonal cycle
changes according to the confidence level. The lowest confidence levels produce composite tracks
with a maximum in summer and a modest low in winter. As the confidence level increases, the
seasonal cycle of cyclone tracks becomes more pronounced. The maximum of cyclone occurrences
shifts to spring and winter and a clear minimum forms in summer. Interestingly, May and October
function as inflection points where results converge regardless of the adopted confidence level. This
suggests that increasing confidence levels may alter the level, but not displace in time, the maximum
and minimum of the seasonal cycle in cyclone tracks. As discussed in section 3.1, cyclones with
higher intensities are mostly expected to take place in winter and spring (Campins et al., 2010;
Flaounas et al., 2015, 2022). Considering thus the direct relationship of pronounced seasonal cycles of
tracks with cyclones intensity, it is plausible to suggest that the bogus tracks which are filtered out due

to increasing confidence levels correspond to the weaker systems in the datasets used to build composite tracks.

- *Cyclone tracks' density:* Figure 13 shows the composite track densities for confidence levels of 2, 4, 7 and 10. All panels in Fig. 13 depict very similar patterns, where most cyclones occur in the western and central Mediterranean Sea surrounding the Italian peninsula. Areas of high track densities are also observed in the lee side of the Alps, over the Aegean Sea, close to Cyprus, in the Black Sea and in Northwest Africa. These favorite locations of Mediterranean cyclogenesis are consistent with the ones found in previous studies, regardless of the atmospheric model or CDTM used to produce the tracks (Lionello et al., 2016; Flaounas et al., 2018; Reale et al., 2022).Given that the seasonal cycle of cyclone occurrence and intensity is proportional to the adopted confidence level and the similar spatial patterns of track densities in Fig. 13, it is plausible that composite tracks of both weak and intense cyclones share the same locations of occurrence. Indeed, when calculating the ratios between track densities of different panels in Fig. 13, we find fairly constant values within the Mediterranean basin with no apparently distinct geographical areas (not shown).

- *Physical characteristics of tracks:* Figure 14a shows that cyclone lifetimes are increasing as a function of the confidence level. A relatively small median of about 30 hours for a confidence level of 2 gradually increases to a median of about 96 hours when considering a confidence level of 10. Such life times are exceptionally long when compared to those for the tracks of individual CDTMs in Fig. 7c. Translation speeds on the other hand are comparably small for all confidence levels, limited to values below 40 km h$^{-1}$. Interestingly, the median of the distributions tends to displace to faster speeds from confidence level 2 to 6. Thereafter, the distribution medians remain rather constant with a value at about 25 km h$^{-1}$. Finally, Fig. 14c shows that the higher the confidence level, the more distinguishable are the intensities of the three cyclone stages. This is consistent with the increase of lifetimes, suggesting that composite tracks of high confidence levels belong to long-lived intense cyclone systems that are plausibly tracked from their early genesis stage until their late decay. This comes in agreement with previous analysis of Lionello et al. (2016) who showed that filtering out weak and slow cyclones improves the agreement among CDTMs.

When comparing confidence levels of 5 and 10 regarding their seasonal cycles (Fig. 12b), intensity distributions (Fig. 14c) and track densities (Figs 13b and 13d), it can be seen that the higher the confidence level, the more intense cyclones tend to concentrate in winter months over the Mediterranean Sea rather than over land areas or close to the mountains where hot spots of track densities are located in Fig. 13b, but also in Fig. 6 for most CDTMs. It is thus plausible for datasets of high confidence levels to tend to capture well-defined, long-lasting cyclones that travel over maritime areas. This is consistent with Pepler et al. (2020) who combined two CDTM methods and showed that pressure lows which were identified by both methods produced higher average rainfall totals than pressure lows which were identified by only a single CDTM (their figure 2). In parallel, tracks of high confidence level tend to neglect weak mountain lows that correspond to long-lasting perturbations of MSLP, or relative vorticity local maxima, produced by several individual CDTMs that plausibly increase the number of bogus tracks.

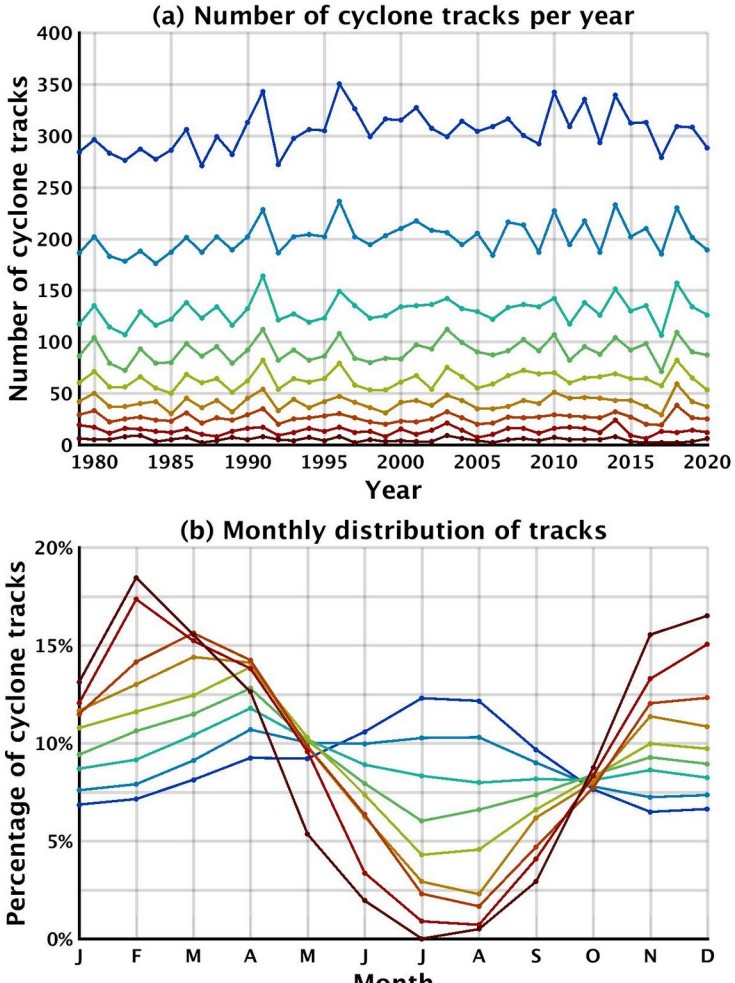

**Figure 12 a** Number of composite cyclone tracks per year when using different confidence levels and an overlapping criterion of 12 hours. Highest number of tracks (line in dark blue) corresponds to the dataset with confidence level of 2 and lowest number of tracks (in dark brown) to the dataset with confidence level of 10 **b** Seasonal cyclone of composite cyclone tracks.

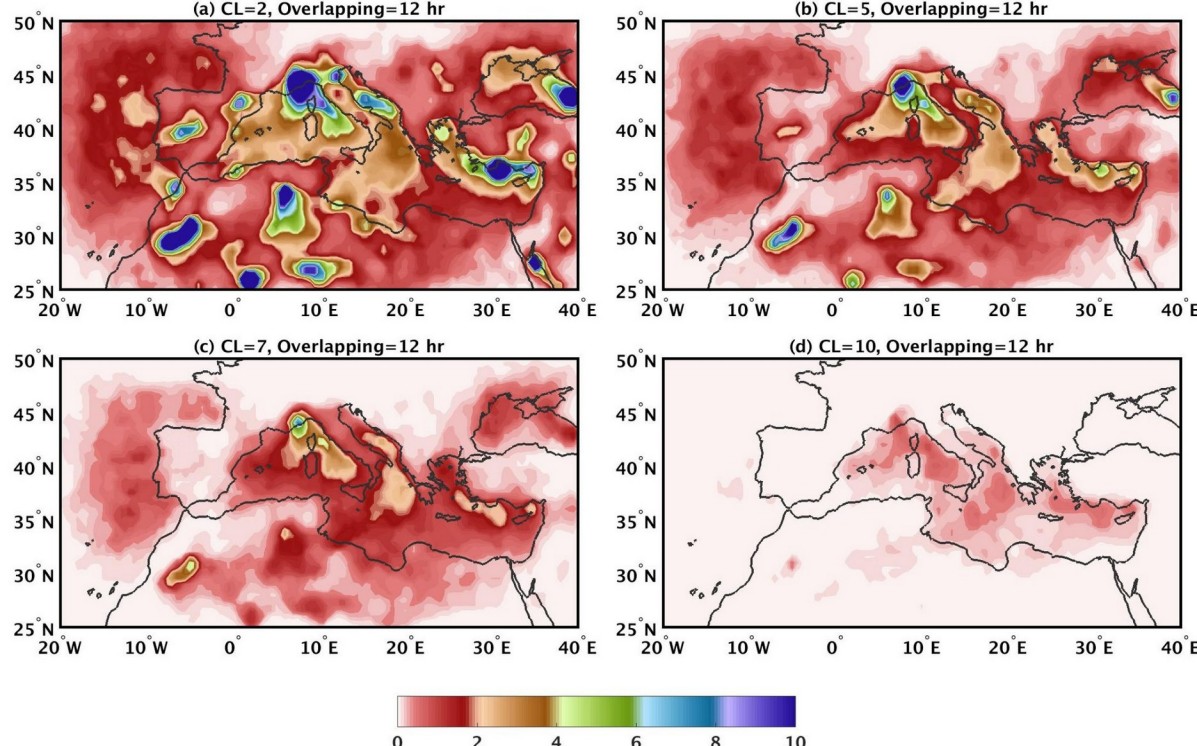


**Figure 13** Spatial density of cyclone tracks for composite tracks of four different confidence levels (CL) and an overlapping criterion of 12 hours. Spatial density expresses the average count of track points per year within a circular area with a radius of $0.5^{o}$.

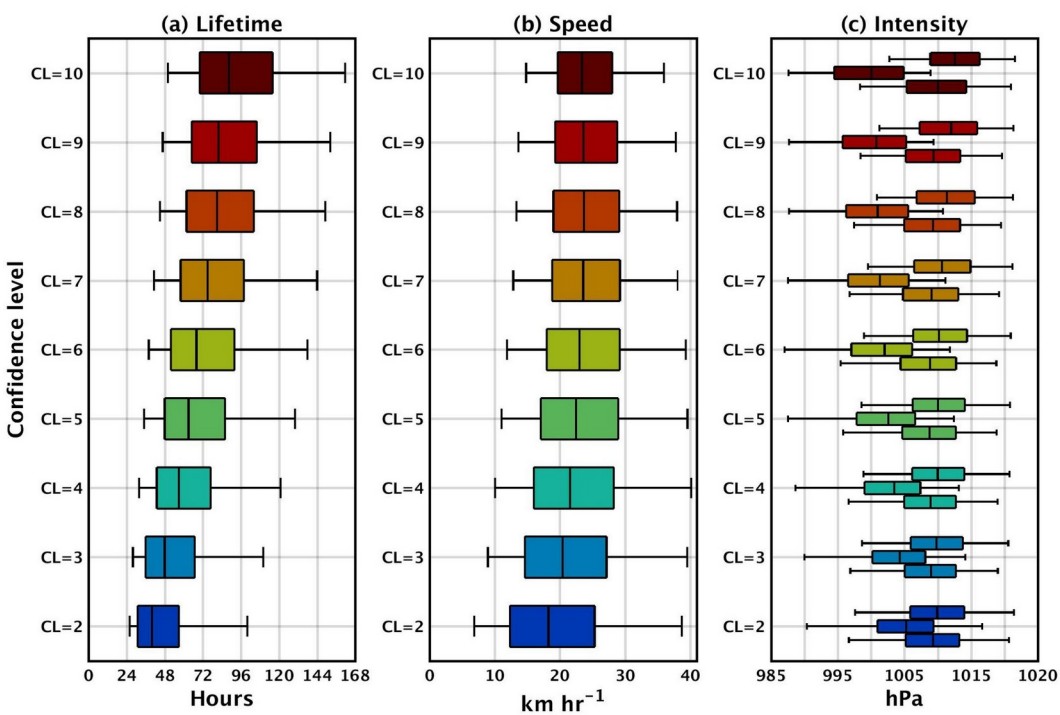


**Figure 14** as in Fig. 7 but for composite tracks of different confidence levels (CL) and an overlapping criterion of 12 hours.

**5. Data availability**

All composite cyclone tracks for different confidence levels are provided as supplementary material in the form of ASCII files. For each confidence level, we provide a separate file that includes a matrix of eight columns and a number of rows that varies among the datasets. Each row corresponds to a single track point, while the eight columns provide the following information:

- Column 1: A cumulatively increasing index that functions as an identifier of unique cyclone tracks.
For instance, all information about the track of cyclone #456 are found in all rows starting with the number 456.

- Column 2: Longitude of track points

- Column 3: Latitude of track points. It is important to note that geographical coordinates are produced using Step 2 of our method and thus may not match the exact location of grid points of
ERA5.

- Column 4: Year of occurrence

- Column 5: Month of occurrence

- Column 6: Day of occurrence

- Column 7: Hour of occurrence

- Column 8: Lowest MSLP value within a 2.5 degrees radius from the geographical coordinates in columns 2 and 3. These values are only meant to function as an approximate reference of intensity. Indeed, geographical coordinates of composite track points in columns 2 and 3 are located in the average location of track points of individual CDTMs. Therefore, values in column 8 may not necessarily correspond to the deepest MSLP, or highest relative vorticity of the tracks.

**6. Discussion and conclusion**

Many CDTMs have been previously used, yielding a substantial amount of climatological findings for extratropical cyclones. However, research has not been able so far to produce reference datasets that would render cyclone climatological results from different CDTMs directly comparable to each other and could be used as a reference to analyze new or updated versions of existing CDTMs. However, it
is rather difficult to produce such best-track datasets for extratropical cyclones. These systems have a complex morphological nature, varying in size and shape, while several centers could be identified within the vicinity of their single meso-to-large scale cyclonic circulation. This complexity is in contrast to tropical cyclones where cyclone systems are clearly distinguishable from their environment and tend to present a single cloudless center. To address this gap in the state-of-the-art in
the field of cyclone tracking, we propose a method that builds datasets of ranked confidence level by combining outputs from 10 different CDTMs. We especially focus on Mediterranean cyclones, which are rather challenging weather systems to track due to their frequent proximity to complex geographical features such as abrupt land-sea transitions and the long mountainous chains that surround the Mediterranean Sea. These geographical features perturb the atmospheric fields which are
used as input datasets to CDTMs. Therefore, CDTMs tend to produce a significant number of bogus tracks that jeopardize the robustness of climatological results. Such bogus tracks might emerge from long lasting field perturbations close to mountains which are "erroneously" perceived as well-organized cyclone systems, or MSLP minima that abruptly develop downstream of a ridge.

In section 2, we described our methodological approach in which composite tracks were produced by
combining the outputs of 10 different CDTMs. The number of combined tracks determines the

confidence level: the more tracking methods agree on the occurrence of a cyclone track, the higher the confidence level of the composite track.

To benchmark CDTMs and composite tracks' performance, five datasets of subjectively tracked cyclones have been produced, each by a different meteorologist. The subjectively tracked cyclones derived from a list of 117 well documented cyclone cases (mostly showing tropical-like characteristics). Considering an overlapping criterion of 24 hours, the five datasets hardly exceeded a similarity score of 80%. This suggests that there can be different human perceptions of the locations and displacements of cyclone centers for a non-negligible amount of cases. Similarity scores between the subjectively tracked cyclones and composite tracks exceeded 80% for confidence levels 2 to 7 (Fig. 10). Thereafter, similarity scores tend to decrease due to the lower probability of consensus among the CDTMs in capturing the same cyclone tracks. In fact, our results suggest that the confidence level acted as a filter that removes weaker, slow and short-lived cyclones. It is also noteworthy that regardless of the corresponding confidence level, our methodological approach produced composite tracks with consistent spatial densities (Fig. 13) and physical characteristics (Fig. 14). Filtering out the weak cyclones, our results show that the higher the confidence level, the more pronounced becomes the seasonal cycle of cyclones occurrence with more systems taking place in winter and spring and fewer in summer.

Proposing composite tracks of high confidence levels as reference datasets has the advantage of tracks concentrating the agreement of many CDTMs. In addition, datasets of high confidence level include more intense systems (Fig. 14) and this reduces the potential of including bogus tracks (i.e. shallow, non-well organized systems). The shortcoming when using datasets of high confidence levels is the likelihood of omitting cyclone tracks that did actually occur but were not "successfully and consistently" tracked by an equal or higher number of CDTMs than the demanded confidence level. Therefore, to propose composite tracks as a reference dataset, one needs to consider a trade-off between "robustness" and "completeness" of the final dataset. In the absence of ground truth on the "correct" number of Mediterranean cyclones and given the fact that similarity scores in Fig. 10 were insensitive to the overlapping criterion, we provide in the supplementary material composite tracks of all confidence levels built with an overlapping criterion of 12 hours.

In this study we used a distance threshold of 300 km to identify similar cyclone track points. This threshold is indeed adapted to the size of Mediterranean cyclones but it could be also envisaged for cyclones in global applications. For instance, let a rather large mid latitude storm be tracked by several CDTMs in a maritime area over the open oceans. In this case, the cyclonic circulation of the storm could encompass an area with a radius of about 1000 km. Very noisy fields due to high spatial resolutions of input fields would potentially lead the CDTMs to find variable representative centers within such a large cyclonic circulation. However, it would be less likely for all, or most CDTMs, to identify these centers in the edges of the cyclonic circulation, such that they are all 300-500 km far from each other. Therefore, a 300-500 km radius would be reasonable to acquire a composite track, still of high confidence level, that omits the CDTMs that identify cyclone centers in the outer areas of the same system.

In the same example, the use of a higher distance threshold (e.g. 1000 km) would produce a composite track point of even higher confidence level that also includes the outlier detected centers. Supposing however that most CDTMs detect cyclone centers relatively close to each other, in both cases of lower and higher distance thresholds, the composite track points would be found in similar average locations. Given the regular splitting and merging of cyclone centers within large cyclonic circulations in the mid-latitudes, a large distance threshold would favor the production of longer composite cyclone tracks and high confidence level. Nevertheless, high distance thresholds would prohibit the early detection of cyclones that start e.g. within frontal areas of large cyclonic circulations. In addition, average track locations would risk to be far from the locations of minima of MSLP, geopotential or maxima of relative vorticity. As a result, our methodological approach is transferable to global or any regional application where a distance threshold of 300 to 500 km would be sufficient to yield meaningful results regardless of the physical characteristics of the tracked

systems. Nevertheless, in any case, the quality and robustness of composite tracks depend on the efficient calibration and number of included CDTMs, respectively.

For a "more general approach" to Mediterranean cyclone climatology, we would recommend the use of datasets with confidence levels of 5 to 7. These confidence levels include a sufficient number of cyclone tracks for climatological studies (50-90 tracks per year) and still retain high similarity scores with subjectively tracked cyclones (Fig. 10). A confidence level of 8-10 would be more appropriate for studies on cyclone dynamics where composite approaches would analyze the most intense systems even if the number of cyclones per year is comparably small to the datasets of other confidence levels.

High similarity scores with subjectively tracked cyclones were also achieved by several individual CDTMs but with much higher number of cyclone tracks per year which were shown to correspond to systems of weaker intensities and shorter lifetimes, with the tracks often presenting indistinct seasonal cycles. Our composite tracks retain high similarity scores but are shown to correspond to deeper cyclones with consistent seasonal cyclones and yield meaningful -less "noisy"- track densities than individual CDTMs (Fig. 6). For these reasons, our methodological approach gives us confidence that it produces adequate reference datasets that include the most possible well-organized systems of cyclonic circulation and the least possible bogus tracks.

## Appendix A

**M01** (Aragão and Porcù 2022): This method was designed to identify and track Mediterranean cyclones taking advantage of the recent availability of a high-resolution reanalysis dataset of ECMWF ERA5. The first step evaluates the Geopotential Height at 1000 hPa (Z1000) of each gridpoint in the Mediterranean region searching for Local Minimums (LM). Then, the list of LM identified at each timestep passes through a filter to keep only the lowest LM within a 5°×5° area, typical extratropical cyclone sizes observed in the region with average values of 500 to 550 km. An additional filter closes the detection step by selecting only LM related to an atmospheric depression with dimensions equivalent to the Rossby-deformation scale (1000 km), applying a Directional Average Spatial Gradients (DASG) of Z1000. In the end, only grid points surrounded by eight positive DASG remained. In the second step, the list of LM is combined using the Nearest Neighbour Method, where the searching box at timestep $t_{n+1}$ was set to a 5°×5° area around the LM position at timestep tn. As the timestep advances, the combined LM positions trace a route that, in turn, ends when it is not possible to find a LM at $t_{n+1}$ inside the search box of the LM at $t_n$, concluding the cyclone lifecycle. Finally, only cyclones lasting more than 24 hours were considered.

**M02** (Flaounas et al., 2014): This is a modified version of the cyclotrack code, based on Flaounas et al. (2014). MSLP is used as input variable instead of relative vorticity at 850 hPa as in the original method. The fields are spatially smoothed using a Gaussian filter with a fixed kernel side of 150 km and a sigma value equal to 2. All cyclone centers are identified as grid points with smaller MSLP values than in the eight surrounding grid points. After cyclone centers are identified, the algorithm starts from the deepest cyclone center in the dataset and produces all of its possible tracks by connecting cyclone centers in consecutive time steps -backwards and forwards in time- provided that they are not 250 km apart. Among all possible tracks, the algorithm chooses the ones that present the least average MSLP difference between its track points. The cyclone centers used to create the track are then discarded and the algorithm continues with building the track of the next deepest cyclone center.

**M03** (Ziv et al., 2015): This routine follows Hewson and Titley (2010), with some modifications that account for irregular cyclone trajectories. The algorithm was designed to have minimal filtering in order to avoid the possible underrepresentation of Eastern Mediterranean cyclones, which are often small and have high minimum pressure relative to other parts of the Mediterranean. Cyclone centers are identified as local MSLP minima in a 15-by-15 grid points window. When two centers are found within a 300 km radius, the shallower one is discarded. In order to connect the centers into tracks, the algorithm calculates a weighted distance metric (scaled by the difference in mid-tropospheric layer

thickness) between each center and its candidate matches in the next time step (t+1). The match with the shortest weighted distance (under 100 km) is added to the track. If no match is found in time step t+1, the search is extended to matches at time step t+2 (t+3 and later centers are not considered). This allows for tracks to have single time step "holes" where the center is not detectable due to noise, topography, etc.


**M04** (Sanchez-Gomez and Somot, 2018): This code is based on the Ayrault (1998) algorithm adapted to the high spatio-temporal resolution of ERA5 and to the peculiarities of the Mediterranean basin. It uses the relative vorticity field at 850 hPa smoothed by averaging values of the ERA5 grid using Gaussian weights over a distance of 225 km. In a first step, the vorticity strongest maxima that

exceeds a threshold of $10^{-4}$ $s^{-1}$ are selected in order to keep only one maximum in a radius of 300 km. When a local vorticity maximum is found (i.e. a cyclone center), a quality criterion based on advection by the wind fields at 850 hPa and 700 hPa and on the vorticity core value is applied to select the matching vorticity maximum at the next time step. This new maximum is kept only if it lies within a range of 150 km from the previous point. At every time step, if a local MSLP minimum is

found in a square of 2.5° side length centered on the relative vorticity maximum, the MSLP location is kept instead of the vorticity point. The trajectories are then validated if they last for longer than 24 hours and if MSLP points are found in the track.

**M05** (Ragone et al., 2018): This method is a slightly modified version of the algorithm used in Ragone et al. (2018), which is partly based on Picornell et al. (2001). First, MSLP fields are spatially

smoothed using a Cressman filter with an influence radius of 200 km. Then, for each minimum, sea level pressure gradient is computed along the eight principal directions inside a circle of radius of 300 km. The pressure minimum is considered as a cyclone center if the maximum sea level pressure gradient along at least 6 directions is larger than 0.5 Pa.km-1. The trajectories are then generated imposing a proximity condition: for each minimum at time t, another minimum at time t+1hr within a

radius of 120km is considered to belong to the same trajectory. **M06** (Picornell et al., 2001; Campins et al., 2006): In this method, a cyclone is defined as a relative minimum in the MSLP field, with a mean pressure gradient greater than or equal to 0.5 hPa per 100 km at least in six of the eight principal directions around the minimum. To avoid excessive noise a Cressman filter is applied. In order to build the cyclone tracks, for each cyclone center the presence of another cyclone center at the next

map is looked for. A searching domain is defined as the elliptical area which extends from the cyclone center along the 700 hPa horizontal wind (considered as the steering level of the movement for the cyclone) and spreads depending on the mean wind speed at this level. If a cyclone center is found into the searching domain, then the two cyclone centers are connected.

**M07** (Hodges, 1994, 1995, adapted from Priestley et al., 2020): Tracking is performed using 850 hPa

relative vorticity as an input variable. Prior to tracking all input data is spectrally filtered to T42 and the influence of planetary-scale waves is removed by masking all wavenumbers less than 5. Tracks are initially identified by searching for vorticity maxima, which are refined using B-spline interpolation and steepest ascent maximization. Cyclones are grouped into tracks using a nearest neighbor approach. Tracks are refined through the minimization of a cost function for track

smoothness, which is subject to adaptive constraints. Tracks must last at least 24 hours and the maximum relative vorticity must exceed 1x10-5 s-1.

**M08** (Lionello et al., 2002; Reale et Lionello, 2013): MSLP is used as input variable. The procedure involves the partitioning of MSLP fields in a certain number of weather systems by identifying sets of steepest descent paths leading to the same minimum, which is a point where the value of MSLP is the

lowest with respect to the eight nearest points. All the points crossed by the same path are assigned to the same cyclone. Moreover small systems which are less than N points far from a deeper system are assigned to the latter. N depends on the resolution of the data and is equal to 20 in the case of ERA5 (that corresponds to a distance between the two cyclone centers of approximately 450 km). The track of the system is then built connecting the position of the cyclone in successive maps.

**M09** (Ullrich et al., 2021; Zarzycki and Ullrich, 2017): MSLP is used as the input variable to the TempestExtremes tracking algorithm. Identification of candidate points requires a minimum in MSLP which must be enclosed by a closed contour of 20 Pa within 1 degree (great circle distance) of the cyclone center. Candidates within 3 degrees of one another are merged with the lower pressure taking precedence. For candidate points to become tracks, the storm must persist for 24 hours, with a maximum gap (time between candidates satisfying the detection criteria) of at most 3 hours. The storm is required to move at least 4 degrees from the start to the end of the trajectory, with maximum distance between candidate points of 2 degrees.

**M10** (Wernli and Schwierz 2006; Sprenger et al., 2017): MSLP is used as input variable. Local MSLP minima are first identified in regions where topography does not exceed 1500 m altitude. Isobars are then identified at a 0.5 hPa interval, and a local MSLP minimum is kept if the isobar enclosing the local MSLP minimum exceeds 100 km in length and is at least 1 hPa higher than the local minimum The resulting set of cyclone centers build then the basis for the cyclone tracking algorithm, which connects cyclone centers at consecutive time steps. To this aim, a search rectangle projected forward in the cyclone's movement direction is used to identify potential candidates. The nearest candidate in the search rectangle is used as the successor. Note that weak cyclones might lack enclosing isobars that meet the criteria. To avoid cyclone tracks being interrupted, two consecutive time steps with no identified cyclone center are allowed.

Appendix B

The list of subjectively tracked cyclones in the supplementary material derives from a series of selected cases that exhibit, -at a certain moment in their lifecycle-, some similarities to weather systems developing in the Tropics. Such similarities refer to the presence of a central cloudless-eye, rainbands extending from the cyclone outskirts, an eyewall, or just intense convection. Thus, the list in the supplementary material includes not only strong cyclones but also rather weak tropical-like cyclones or subtropical storms. The conditions for the inclusion of cyclones in the list were mainly based on visual identification rather than on objective identification of physical or thermodynamic properties.

More precisely, the 117 cyclone cases originate from reviewing the scientific literature (e.g. articles that include the analysis of individual case studies, or provide a list of different cases such as in Nastos et al., 2017 and Tous and Romero, 2013), as well as from Wikipedia articles on specific weather events. Moreover, in our list, we include cases from the extensive collection work of the meteorology group of the University of the Balearic Islands. Although updated up to 2008, the website http://meteorologia.uib. eu/medicanes/medicanes_list.html still contains several Medicanes and depressions. Another important source of cyclone cases included in our list is the website www.medicanes.altervista.org, maintained by Daniele Bianchino. This website includes till 2021 a very detailed list of Mediterranean tropical disturbances/depressions/storms and hurricanes (depending on estimated wind speed strength).

Figure B1 shows the tracks of all subjectively tracked cyclones. The grand majority of these tracks over maritime areas, in particular in the Central and Western Mediterranean. Nevertheless, tracks are not repeated identically in all five datasets. Indeed, 37 out of 117 cyclones were included in all five of them, 75 cyclones were included in at least three of them, while 13 cyclone cases from the 117 in the list were not included in any dataset. Table B1 summarizes the subjective detection rate of the 117 cyclones. The cases not included in any dataset are mostly related to shallow low-pressure systems. The average duration of subjectively tracked cyclones is 37, 40, 54, 37 and 48 hours for datasets D01, D02, D03, D04 and D05 respectively. Such lifetimes are indeed comparable to the ones produced by different CDTMs in Fig. 7a. Finally, cyclone intensity distributions are shown in Fig. B2. They are also comparable to the ones observed in Fig. 7b although mature stages reach deeper MSLP values than those observed in the distributions based on objectively tracked systems.

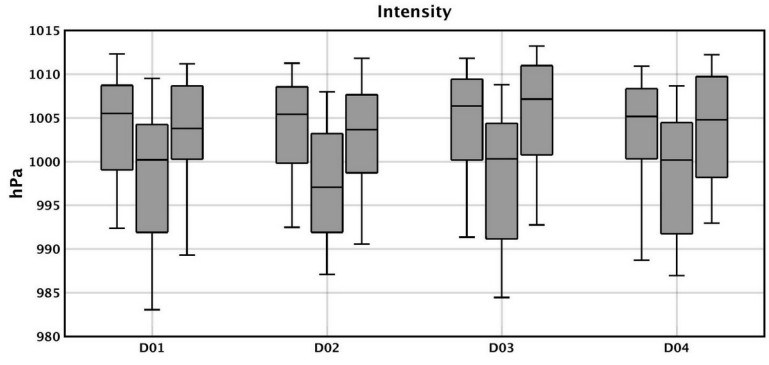

Figure B1 Tracks in the five datasets of subjectively tracked cyclones. Colors are produced randomly to ease readability.

Figure B2 Boxplots showing the distribution of cyclone intensities -measured by core MSLP- for each dataset of subjectively tracked cyclones. Three boxplots are shown for each dataset: the lower one shows the distribution of MSLP at cyclone first track points, the middle one shows the distribution of MSLP at the time of lowest MSLP and the upper one shows the distribution of MSLP at cyclone last track points. For all boxplots, the boxes depict the 25th, 50th and 75th percentiles, while whiskers depict the 5th and 95th percentiles.

| Number of cyclone cases | 13 | 12 | 17 | 14 | 24 | 37 |
|---|---|---|---|---|---|---|
| Number of subjective detections | 0 | 1 | 2 | 3 | 4 | 5 |

Table B1 Summary of subjective cyclone tracking of the 117 selected cases that took place in the period 1979-2020 (provided as supplementary material) by five meteorologists.

## Author contribution

EF conceptualized the composite tracks approach, developed the methodology, performed the analysis and wrote the initial draft. LA, LB, BD, AK, JK, MAP, MDKP, MR, MR, DS and MS provided ERA5 cyclone tracks. EF, LA, SD, PP and ES each produced a dataset of subjectively tracked cyclones using a MatLab program written by EF. The program was applied to 117 past cyclone cases, all derived from a list that was made available and updated by MMM. This study is the outcome of collaborative work between all co-authors in the framework of MedCyclones COST Action (CA19109). Meetings were held on a regular basis from April 2021 until the submission of this paper. All co-authors participated in preparing the final draft of the manuscript, providing valuable comments, reviews and edits.

## Acknowledgements

This study is based upon work from COST Action MedCyclones (CA19109), supported by COST (European Cooperation in Science and Technology), www.cost.eu. The authors thank ECMWF and Copernicus for making the ERA5 dataset available. EF has been supported by the Stavros Niarchos Foundation (SNF) and the Hellenic Foundation for Research and Innovation (H.F.R.I.) under the 5th Call of "Science and Society" Action – "Always Strive for Excellence –Theodore Papazoglou" (Project Number: 7269). BD has been supported by Région Occitanie and Météo-France. MAP has been supported by Ministerio de Ciencia e Innovación – Agencia Estatal de Investigación (Spain) through the Project TRAMPAS (PID2020-113036RB-I00 / AEI / 10.13039/501100011033). MR has been supported in this work by OGS and CINECA under HPC-TRES award number 2015-07 and by the project FAIRSEA (Fisheries in the Adriatic Region - a Shared Ecosystem. Approach) funded by the 2014 - 2020 Interreg V-A Italy-Croatia CBC Programme (Standard project ID 10046951). MJR was supported by the Met Office Hadley Centre Climate Programme funded by BEIS and Defra (GA01101).

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

| Code | Main references for method description | Variable used to identify cyclone centers |
| --- | --- | --- |
| **M01** | Aragão and Porcù (2022) | Geopotential Height at 1000 hPa |
| **M02** | Flaounas et al. (2014) | MSLP |

| | | |
|---|---|---|
| **M03** | Ziv et al. (2015) | MSLP |
| **M04** | Ayrault, (1998); Sanchez-Gomez and Somot, (2018) | Relative vorticity field at 850 hPa and MSLP |
| **M05** | Ragone et al. (2018) | MSLP |
| **M06** | Picornell et al. (2001); Campins et al., (2006) | MSLP |
| **M07** | Hodges (1994, 1995), as applied in Priestley et al. (2020) | Relative vorticity field at 850 hPa |
| **M08** | Lionello et al. (2002); Reale et Lionello (2013) | MSLP |
| **M09** | Ullrich et al. (2021); Zarzycki and Ullrich (2017) | MSLP |
| **M10** | Wernli and Schwierz (2006); Sprenger et al. (2017) | MSLP |

**Table 1** The code name, references and input variable of the 10 different CDTMs used in this study, described in more detail in the Appendix.

1025