# Peer review of "A composite approach to produce reference datasets for extratropical cyclone tracks: Application to Mediterranean cyclones"

_Weather and Climate Dynamics, 2022_

## Referee Comment (RC2)

**Review of "A composite approach to produce reference datasets for extratropical cyclone tracks: Application to Mediterranean cyclones"**

February 9, 2023

This manuscript presents a new approach which combines tracks from individual cyclone tracking algorithms to produce composite / best estimate tracks. Overall the manuscript is interesting and well written and the resultant data sets will be of use to the scientific community. I have three main concerns: (1) how applicable this method is / how necessarily this method is to extra-tropical cyclones in other geographic regions, (2) how physically consistent the resultant composite tracks are and (3) how the benchmark data set was created. These are discussed in more detail below along with some other minor comments:

**1 Major Comments**

1. Applicability / relation to extra-tropical cyclones in other geographic regions.

    (a) The manuscript discusses extra-tropical cyclones and Mediterranean cyclones almost inter-changeably without acknowledging that Mediterranean cyclones are a special subset of extra-tropical cyclones. Better discussion of how Mediterranean cyclones differ from e.g. North Atlantic cyclones should be included in the introduction. This is important because certain characteristics of Mediterranean cyclones make them much harder to track than extra-tropical cyclones in the major storm track regions which raises the question – is this method of composite tracks necessarily in all locations?

    (b) The manuscript does not clearly acknowledge that many (although not all) of the tracking algorithms were first developed primarily to identify synoptic-scale cyclones in the major oceanic storm tracks and as such, were not originally designed to be used to identify Mediterranean cyclones. This should be acknowledged and it should be made clearer in the main body of the text which tracking algorithms were specifically designed to identify Mediterranean cyclones.

(c) Are bogus tracks a much bigger problem in the Mediterranean than in the North Atlantic / Pacific / Southern Hemisphere storm tracks? e.g. is this largely a Mediterranean problem? These are mentioned frequently throughout the manuscript yet no evidence of these are given. Could some examples be included or at least some evidence to show a reader that these really do exist and are a significant problem? In addition, it is my understanding that by default many algorithms tracks many cyclones but then filtering is performed to remove weak / stationary cyclones. Does this pre-existing filtering in many methods remove the bogus tracks?

2. Physical consistency. A major question I have regarding this method is, is the final produced composite track still physically consistent with the spatial pattern and time evolution of the input data (e.g. with the MSLP data from ERA5)? I would very much like to see some evidence of this in the manuscript e.g. plot the cyclone centre from the composite track and the mean sea level pressure at multiple time steps and see if it looks as physically plausible as one individual track.

3. Generation of the benchmark data set

(a) Line 247. Selection of the 120 cyclones that manual subjective tracks were created for. There is not enough details of how these cyclones were selected nor of these cyclones' properties (life time, intensity) relative to all cyclones. It is stated that these cyclones were selected based on previous case studies. As case studies tend to only be performed on extreme / unusual storms, this suggests that these 120 selected cases are not representative of the whole data set.

(b) Line 249 – 255. Information given to meteorologists to identify the tracks manually. I think this approach is somewhat flawed as the meteorologists have only been given the MSLP field which is either the same or less information than the tracking algorithms have. This approach wants the meteorologists to behave like the automated methods, rather than the intelligent meteorologists that they are. In my opinion, the meteorologist should have been allowed many more fields / information to create a true bench-mark data set that really identifies true cyclone tracks (i.e in the way a forecaster would), not only those cyclone tracks which are evident in MSLP.

**2 Minor Comments**

1. Table 1. Could any additional filtering applied to the tracks (e.g. length or duration of the track) be added to this table. It would better allow a reader to see how the methods differ.

2. Line 149, Appendix L618. In the text (line 149) it is written that only cyclone tracks which last for at least 24 hours are retained. When checking the Priestley reference in

line 618, it appears that M07 only keeps tracks which last 48 hours (and travel 1000 km). Please resolve this confusion (based on Figure 6 I think tracks that last at least 24 hours are retained). Adding more details to the appendix for M07 would resolve this. It also appears that M07 only keeps tracks if the maximum vorticity exceeds a given threshold. This information should also be added.

3. Line 161 – 162. The distance criteria for whether tracks overlap or not is given here as 300 km. This threshold value is justified based on the scale of Mediterranean cyclones. Therefore, would this value need to be different if this method was used for e.g. north Atlantic cyclones which are generally larger but potentially easier to track? Aspects of this new method of combining tracks which are specifically for Mediterranean cyclones need to be more clearly highlighted.

4. Line 200 / Figure 2, panels e and f. These panels are hard to read. Would it be possible to only show the composite track / black line and the individual tracks for each tracking algorithm? i.e. the black dots of varying sizes could be removed.

5. Line 307. In my opinion M06 also has a prominent seasonal cycle (albeit with a different phase to M04 and M09) and should be noted here.

6. Line 313 / track density. How was track density computed and is it the same for all 10 methods? Please add these details. Related to this is in Figure 5 the track density goes to the edge of the domain for some methods (e.g. 4, 7, 9) whereas for others (1,3,6,8) there is some halo / gap around the edge. Why is this?

7. Figure 5. There is some overlap with the longitude labels on the bottom right of each panel. Could this be fixed?

8. Line 333 – 334. This is a confusing and redundant sentence. It is not surprising that the lowest values occur at the time of minimum MSLP. In addition, "mature" stage is not clearly defined and could be misunderstood to mean occluded / after the point of minimum pressure.

9. Figure 6. To be consistent with the other figures, could method 01 be at the top and M10 at the bottom?

10. Line 508: Suggest replacing "extra-tropical cyclones" with "Mediterranean cyclones" here.

---

## Author Response (AR1)

Author responses to Reviewers comments for the manuscript:

**"A composite approach to produce reference datasets for extratropical cyclone tracks: Application to Mediterranean cyclones"**

April, 2023

**Reviewer #1**

*Review of "A composite approach to produce reference datasets for extratropical cyclone tracks: Application to Mediterranean cyclones"*

*I enjoyed reading this interesting and well-written paper. The authors use a novel approach to combining multiple cyclone detection and tracking schemes, which aims to identify those Mediterranean cyclones that are consistently detected between methods. They also evaluate this using a combined dataset from 5 subjective analyses, demonstrating the uncertainty inherent in subjective tracking datasets. I have a range of relatively minor comments and suggestions that I hope will improve an already very good paper.*

We would like to thank Reviewer #1 for the positive review and for providing insightful comments that greatly improved our work.

**Specific comments:**

*1. Colour schemes - many of the figures in your paper use colour schemes that include both red and green and are not colourblind-friendly. While I acknowledge that finding 10 unique colours for the 10 methods in e.g. Figure 4 is difficult, the authors could at least use viridis or a diverging red-blue colourbar for Figures like 5 and 8.*

We thank Reviewer #1 for the thoughtful suggestion. We tried our best to change colours following "Aid in color blindness color tables" from NCL Graphics (see link below) for almost all figures.

(https://www.ncl.ucar.edu/Document/Graphics/color_table_gallery.shtml#Aid_in_color_blindness)

*2. While the focus of the paper is on the objective tracking results, I would love to see a bit more analysis of the subjective tracking, as there may be some additional insights in your subjective dataset that will be useful for future research to draw on. What proportion of the 120 cyclones were identified by all 5 experts? What proportion were identified by none? If you designed your dataset based on historical case studies I assume it tends to include medicanes with major impacts - does the fact that many of these are not identified by experts indicate that your duration criteria is too restrictive for studies that want to identify impactful events? How were the case studies distributed throughout the year, and was the matching between the experts better in winter?*
*Supplementary Material 1 mentioned at L247 does not seem to be currently available, but I hope it includes a summary table listing the dates of the 120 events and how many of the experts identified them.*

We thank the Reviewer #1 for this comment. Indeed, most of our 120 cases (actually 117 in the period 1979-2020) correspond to Medicanes or cyclones with tropical-like features. Most cases have been discussed in scientific works and few of them were proposed as such by media reports.

In approximately all known Medicanes, lifetime is greater than 24-hours (from cyclogenesis to decay stage), albeit the formation of a cloudless "eye" -that characterizes medicanes- is a short term feature. Since CDTMs use physical fields as input data, the cloudless "eye" criterion has no effect on the cyclone tracking procedure. In order to accommodate with the Reviewer's comment we now include an Appendix B in the manuscript where manual tracking procedures, the nature of the 117 selected cases and the agreement among the five datasets are thoroughly discussed:

"*Appendix B*

*The list of subjectively tracked cyclones in the supplementary material derives from a series of selected cases that exhibit, -at a certain moment in their lifecycle-, some similarities to weather systems developing in the Tropics. Such similarities refer to the presence of a central cloudless-eye, rainbands extending from the cyclone outskirts, an eyewall, or just intense convection. Thus, the list in the supplementary material includes not only strong cyclones but also rather weak tropical-like cyclones or subtropical storms. The conditions for the inclusion of cyclones in the list were mainly based on visual identification rather than on objective identification of physical or thermodynamic properties.*

*More precisely, the 117 cyclone cases originate from reviewing the scientific literature (e.g. articles that include the analysis of individual case studies, or provide a list of different cases such as in Nastos et al., 2017 and Tous and Romero, 2013), as well as from Wikipedia articles on specific weather events. Moreover, in our list, we include cases from the extensive collection work of the meteorology group of the University of the Balearic Islands. Although updated up to 2008, the website http://meteorologia.uib. eu/medicanes/medicanes_list.html still contains several Medicanes and depressions. Another important source of cyclone cases included in our list is the website www.medicanes.altervista.org, maintained by Daniele Bianchino. This website includes till 2021 a very detailed list of Mediterranean tropical disturbances/depressions/storms and hurricanes (depending on estimated wind speed strength).*

*Figure B1 shows the tracks of all subjectively tracked cyclones. The grand majority of these tracks over maritime areas, in particular in the Central and Western Mediterranean. Nevertheless, tracks are not repeated identically in all five datasets. Indeed, 37 out of 117 cyclones were included in all five of them, 75 cyclones were included in at least three of them, while 13 cyclone cases from the 117 in the list were not included in any dataset. Table B1 summarizes the subjective detection rate of the 117 cyclones. The cases not included in any dataset are mostly related to shallow low-pressure systems. The average duration of subjectively tracked cyclones is 37, 40, 54, 37 and 48 hours for datasets D01, D02, D03, D04 and D05 respectively. Such lifetimes are indeed comparable to the ones produced by different CDTMs in Fig. 7a. Finally, cyclone intensity distributions are shown in Fig. B2. They are also comparable to the ones observed in Fig. 7b although mature stages reach deeper MSLP values than those observed in the distributions based on objectively tracked systems.*"

**(a) Dataset D01**

**(b) Dataset D02**

**(c) Dataset D03**

**(d) Dataset D04**

**(e) Dataset D05**

**Figure B1** *Tracks in the five datasets of subjectively tracked cyclones. Colors are produced randomly to ease readability.*

[Figure]

**Figure B2** *Boxplots showing the distribution of cyclone intensities -measured by core MSLP- for each dataset of subjectively tracked cyclones. Three boxplots are shown for each dataset: the lower one shows the distribution of MSLP at cyclone first track points, the middle one shows the distribution of*

*MSLP at the time of lowest MSLP and the upper one shows the distribution of MSLP at cyclone last track points. For all boxplots, the boxes depict the 25th, 50th and 75th percentiles, while whiskers depict the 5th and 95th percentiles.*

| Number of cyclone cases | 13 | 12 | 17 | 14 | 24 | 37 |
|---|---|---|---|---|---|---|
| Number of subjective detections | 0 | 1 | 2 | 3 | 4 | 5 |

**Table B1** *Summary of subjective cyclone tracking of the 117 selected cases that took place in the period 1979-2020 (provided as supplementary material) by five meteorologists.*

*3. At L310 you attribute the varying seasonal cycles to weaker cyclones being more common in the summer months. If that were the case I would expect to see a correlation between the total number of cyclones identified by a method and the proportion of lows in the summer months, but that does not seem to be the case - M07 has the second-highest frequency of lows and M08 has the second-lowest, but they both seem to have very uniform seasonal distributions. I think this needs further assessment, as there may be some more complex factors at play e.g. related to the spatial distributions of lows in different methods.*

We thank the Reviewer #1 for this insightful comment. In order to better accommodate with the Reviewer's comment we now include in Fig. 5 (Fig. 4 in original submission) an additional panel that shows an overall improvement among the tracking methods' seasonal cycle for the most intense systems. We reformulate the paragraph as follows:

*- Interannual and seasonal cycle: Figure 5a shows that the number of tracks differs significantly among the methods, with M03 having the largest number of detected cyclones (about 500 per year). On the other hand, methods M04, M08 and M10 show the lowest number of tracks (approximately 100 to 120 per year). Moreover, there is no clear interannual trend for either method while only a few anomalous years are observed. Figure 5b shows that the seasonal cycle is fairly weak for all the CDTMs. Cyclogenesis tends to be evenly distributed along the year with 7 to 12% per month. Methods M04 and M09 show prominent seasonal cycles with a minimum (maximum) in (summer) spring. On the other hand, several methods (M01, M03, M06 and M10) show a peak of cyclogenesis during the summer months of the year. Despite these differences, when considering only the 500 most intense cyclones in each method, seasonal cycles of cyclogenesis become more similar. Indeed, Fig. 5c shows a prominent seasonal cycle that comes in agreement with recent studies on Mediterranean cyclones climatology where most intense cyclones tend to occur in winter and spring (e.g Campins et al., 2011; Flaounas et al., 2015; Lionello et al., 2016). The comparison of Figs 5b and 5c suggests that it is rather "easier" for CTDMs to agree on the seasonal cycle of well-developed intense cyclones than on that of shallow cyclones that are expected to be less distinguishable from bogus tracks making their identification more uncertain. For instance, let a simplistic approach where cyclone centers are defined as grid points of the lowest MSLP among their eight neighboring ones. If the identified centers*

*correspond to very low values (e.g. 980 hPa), then it is plausible to assume that these centers are related to rather deep cyclones. On the other hand, if centers are close to the regional average (e.g. 1015 hPa), then they likely correspond to rather weak or shallow systems, or simply to local minima of MSLP that are not related to well formed cyclonic circulation, i.e. to a bogus track.*

[Figure]

**Figure 5 a** Number of cyclone tracks per year for each CDTM **b** Average monthly distribution of cyclone tracks occurrence for each CDTM with respect to the total number of cyclone tracks per year **c** Monthly distribution of 500 deepest cyclones for each CDTM.

*4. Section 3.2/Figure 7 - I find it interesting and surprising that the similarity scores in Figure 7 seem to be uncorrelated with the total number of lows each method generates in Figure 4 - I would have expected M07 to perform a lot better given its high track frequency. Do you have any explanation of this?*

We thank the Reviewer #1 for pointing out this point. First we would like to stress that the number of subjectively tracked cyclones is small compared to the expected number of systems in a 42-year climatology. Therefore, Figure 7 could hardly provide insights into the skills of the methods in tracking Mediterranean cyclones. Nevertheless, it is plausible to expect that CDTMs producing a high number of cyclone tracks are also able to capture the few subjectively tracked systems. Given the many different criteria used in the 10 CTDMs for identifying and tracking cyclone centers, it is rather difficult to pinpoint the reasons why several CDTMs have been able to capture the cases reported in the five reference datasets and others not. It is however important to stress that several of the methods used in this study were developed and substantially tested for the Mediterranean region. This potentially allows certain CDTMs to show better performances in Fig. 7 . M07 is the only one that uses relative vorticity as an input field while the manual tracks were identified using MSLP. This different approach in cyclone detection is also expected to have an impact on the results of Fig. 7. To address the Reviewer's comment we added the following sentence in section 3.2:

*"It is noteworthy that several CDTMs were developed and substantially tested for studies in the Mediterranean region (Appendix A). This could result in a more favorable calibration for Mediterranean cyclones. Especially concerning the performance of M07, this is the only CDTM that uses relative vorticity as an input field while the manual tracks were identified using MSLP. This inconsistency may also have an impact on the results of Fig. 7."*

*5. Figure 9 would be more useful I think if it showed the "hit rate" (the proportion of subjective lows detected), so that the denominator is the same for the whole plot.*

This is indeed the hit rate. The caption is now corrected:

*"…. Similarity scores are defined as the number of similar tracks in two datasets divided by the number of subjectively tracked cyclones..."*

*6. L462 - Please share the correlation values - Figures 11c and d look very different to my eye, as Fig 11d has almost no lows in the Atlantic or the Black Sea but in Figure 11c the numbers in those regions are only slightly lower than in the Mediterranean sea. If the correlations are only calculated over a smaller subregion e.g. the Mediterranean sea, maybe show a contour around that area in Figure 11.*

We thank the Reviewer #1 for this comment. The figure below shows track densities as normalized index with respect to the maximum and minimum of track counts within 0.5 degrees of all grid points. Hotspots of cyclone activity are similar in the four panels, nevertheless we agree with the Reviewer that correlation scores might not be as high as 1. Originally in the paragraph we meant that correlations are very high when comparing track densities of consecutive confidence levels (e.g. comparing datasets with CL=2 and CL=3 or CL=7 and CL=8). In order to avoid confusion we have removed this sentence .

[Figure]

7. Comparing Figure 10a and Figure 4, it becomes obvious that because the numbers of cyclones vary significantly between datasets, lows identified using 2 or 3 methods will be dominated by agreement between just the subset of methods with high frequencies (M03,M07,M06 and M01). The paper would benefit from a figure that tries to quantify this, to understand which methods are most responsible for determining the seasonality/spatial patterns of the combined tracks that are then obvious in Figures 10b and 11. I would imagine something like Figure 8, showing that e.g. for a confidence level of 2 (fake numbers), 60% of composite tracks include M03 but only 20% include M08, since M08 is less common. This increases with confidence level, so at level 10 each method is included 100% of the time, by definition. Similarly, a plot showing what proportion of all tracks for a method are included in the combined dataset - e.g. that a confidence level of 8 includes 50% of M08 tracks but only 5% of M03 tracks. Bonus if this could be shown for different months or seasons.

We thank the Reviewer for the suggestion. The confidence level is computed taking into account the minimum number of CDTMs that detect a specific track. For instance, a dataset with confidence level equal to 2 includes tracks that were detected by at least 2 CDTMs. Therefore, datasets of high confidence levels are subsets of other datasets built with lower confidence levels. In order to address the Reviewer's comment, we now clearly stress that in the text.

*"The number of methods used to create a composite track is measured by a "confidence level", which ranges from 2 to 10. For instance, a confidence level equal to 5 suggests that a track point was captured by at least five CDTMs. As such, final composite track datasets of any confidence level are always subsets of datasets with lower confidence level."*

In addition, we have produced the following figure and updated the text in Section 3.2 as follows:

*"In order to provide further insights into the contribution of each individual CDTM to the resulting composite tracks, Fig. 11 shows the contribution (in %) of each CDTM to composite tracks (using the 12 hours criterion). For instance, 72% of composite tracks of confidence level 6 were produced with contributions from M06. By design, the composite tracks of confidence level 10 require the participation of all CDTMs. Therefore, contributions of each CDTM to the highest confidence level is by default 100%. On the other hand, confidence level 2 demands only two CDTMs to agree on the detection of a specific track. This naturally yields lowest percentages for all CDTMs in Fig. 11 with M04, M08 and M10 contributing to the composite tracks of confidence level 2 only by 27%. In fact, it is more likely for a low number of sensitive CDTMs to detect shallow or weak cyclonic systems whereas another CDTM would fail if it applied more strict criteria. M01 and M03 contribute to more than half of composite tracks even in datasets with low confidence level. Beyond confidence level 5, all CDTMs contribute to the majority (>50%) of composite tracks with an average contribution of 85% to composite tracks of confidence level 7. Finally, results in Fig. 11 seem to be independent of the season although contributions tend to be slightly smaller (higher) in summer (winter) for confidence levels less than 4 (not shown)."*

[Figure]

*Figure 11* *Contribution of CDTMs to the production of composite tracks using the 12 hours criterion.*

As discussed in the text right above, Fig. 11 shows no particular seasonal dependence and therefore this issue was not further discussed in the manuscript. The following figure provides the same results as in Fig. 11 right above but for tracks in Winter (DJF), Spring (MAM), Summer (JJA) and Autumn (SON).

[Figure]

8. *Conclusions - I think the conclusions would benefit from some discussion of potential extensions/applications. Which confidence level do you think would be most applicable to identifying medicanes with significant impacts e.g. rain, given the tendency to favour long-lived events meant that many of the case study events were not identified by your subjective analysis? How applicable do you think this approach would be for cyclones in other areas or globally?*

We thank the Reviewer for the suggestion. Conclusions have been now enriched with a discussion on the applicability of our method to extratropical cyclones beyond the Mediterranean region (please see our reply to Reviewer #2 to major comment 1 ).

Regarding the first part of the Reviewer's comment, there is no commonly accepted physical definition of Medicanes. Although they typically correspond to intense systems, Medicanes are not necessarily the most intense cyclones that develop in the Mediterranean region. In fact, high-impact weather events might emerge from rather strong and deep systems such

as Medicanes (Nastos et al., 2017) as well as from small and shallow vortices (e.g. Miglietta et al., 2023). The purpose of building and using datasets of high confidence level is related to the removal of large portions of bogus tracks and retaining at the same time well-formed cyclones. In this sense, long-lasting and highly impact events are expected to be more intense than others and therefore to be better captured in high confidence level datasets. On the other hand, a confidence level of 2 is expected to include systems with shorter lifetimes but still yields a median of 36 hours (Fig. 12a of the original submission) which anyway is not considered as a short lifetime for the standards of Mediterranean cyclones (Lionello et al., 2016).

Miglietta, M.M., Buscemi, F., Dafis, S., Papa, A., Tiesi, A., Conte, D., et al. (2023) A high-impact meso-beta vortex in the Adriatic Sea. Quarterly Journal of the Royal Meteorological Society, 149( 751), 637– 656. Available from: https://doi.org/10.1002/qj.4432

*Minor comments:*
*9. I don't really like the term "bogus tracks" - it implies that the tracks that are only detected by some methods are wrong/bad, when some of those may indeed be real cyclones that caused real impacts. I'm not sure of a better term to use, but it's something to consider when you discuss them.*

As stated in the abstract, with "bogus tracks" we refer to method artifacts, i.e. to tracked features that can be hardly perceived as well-formed cyclones. In order to better explain this point we revised the introduction as follows:

*"… On the other hand, less strict criteria produce a large number of "bogus tracks", which can be perceived as "false positives", or more generally as CDTM artifacts. Bogus tracks might correspond to persistent weak MSLP perturbations or long-lasting vorticity local maxima due to abrupt wind steering (e.g. close to steep topographic features). More generally they can hardly be interpreted as well-organized vortices."*

*10. Figure 2d/L207 - I assume that the track components that were rejected from this composite (i.e. the red dots) remain in the pool of data, so the red dots end up being a second track in the dataset with confidence ~5?*

All track segments not contributing to the composite tracks are discarded (please also see our reply to the Reviewer#2 major comment 2). In order to accommodate with the Reviewer's comment the following sentence has been added in section 2.1: :

*"It is noteworthy that all tracks found to be similar in step 2 are discarded even if they did not contribute to the composite tracks. As a result, the omitted ensemble of tracks in the eastern Mediterranean (Fig. 2f) were not later used to produce a new composite track of lower confidence level."*

*11. L373 - Do you have any explanation of why M01-M02 and M01-M08 have stronger similarities, e.g. linked to characteristics of the methods?*

The methods listed by the Reviewer differ significantly among them, from the identification of local minima to the tracking procedure in each timestep. For instance, M02 and M08 use

MSLP as input field, however M01 uses geopotential height at 1000hPa. In fact, significant divergences among the methods can be noticed on the total number (Fig. 4a), lifetime and displacement speed (Fig. 6) of their tracked systems. Such differences make associating the high similarity scores (Fig. 8) with the characteristics of each method extremely difficult.

12. In most figures, the methods are ordered from M01 at the top to M10 at the bottom, but in Figure 6 M10 is at the top. It would be good for the order to be consistent.

We thank the Reviewer for spotting this error in the figure. All figures are now consistent on the order of methods.

13. The authors may want to cite Pepler et al. (2020), who combined two CDTM methods and showed that lows identified by both methods produced higher average rainfall totals than lows identified by only a single CDTM (their Figure 2).

We thank the Reviewer for this suggestion. We included this information in section 4:

"It is thus plausible for datasets of high confidence levels to tend to capture well-defined, long-lasting cyclones that travel over maritime areas. This is consistent with Pepler et al. (2020) who combined two CDTM methods and showed that pressure lows which were identified by both methods produced higher average rainfall totals than pressure lows which were identified by only a single CDTM (their figure 2). In parallel,…."

14. I am surprised to see that the method of Kouroutzoglou (2011 etc) is not included in this paper, given that it was used for several key papers on medicane climatology and J Kouroutzoglou and H Flocas are both authors of this paper. Is there a reason for this?

The MS method (Murray and Simmonds 1991) was indeed employed for many climatological studies related to explosive cyclones and cyclonic tracks in the Mediterranean with spatial resolution down to 0.5° degrees and 6-hourly time intervals. In the context of this study, we have initially included the MS method along with the other methods. However, our test runs met technical difficulties in identifying intense cyclones in higher spatial (0.25 degrees) and hourly time intervals. After several sensitivity tests and preliminary results, we discarded this method from our work.

Murray, R. J., and I. Simmonds, 1991a: A numerical scheme for tracking cyclone centres from digital data. Part I: Development and operation of the scheme. Aust. Meteor. Mag., 39 , 155–166.

15. The insight from lines 282-285 - that a similarity of 80% is as good as you can get even between subjective methods - is really interesting. I think it should be highlighted in the conclusions.

Thank you for this comment. In order to address the Reviewer's comment the following paragraph has been added in the conclusions:

"To benchmark CDTMs and composite tracks' performance, five datasets of subjectively tracked cyclones have been produced, each by a different meteorologist. The subjectively

*tracked cyclones derived from a list of 117 well documented cyclone cases (mostly showing tropical-like characteristics). Considering an overlapping criterion of 24 hours, the five datasets hardly exceeded a similarity score of 80%. This suggests that there can be different human perceptions of the locations and displacements of cyclone centers for a non-negligible amount of cases."*

*Technical comments:*
*L169 etc - "Exemplary" is generally used to mean "very good". Unless you mean to say that the cases in Figure 2 are two of the best in the whole dataset maybe you should call them "cases" or "examples".*

"Exemplary" has been removed or replaced with "cyclone cases".

*L628 - What is N for the ERA5 data?*

We apologize with the Reviewer because we did not provide this additional information in the text. As discussed in text "N" is dependent on the resolution of the input and in particular on the long-lat spacing. In the case of ERA5, N is equal to 20 which corresponds to a distance between the two weather systems of approximately 450 km (444 km to be more precise). In order to address the Reviewer comment the sentence is slightly modified as :

*"Moreover small systems which are less than N points far from a deeper system are assigned to the latter. N depends on the resolution of the data and is equal to 20 in the case of ERA5 (that corresponds to a distance between the two cyclone centers of approximately 450 km)."*

*References:*
*Kouroutzoglou, J., Flocas, H. A., Keay, K., Simmonds, I., and Hatzaki, M.: Climatological aspects of explosive cyclones in the Mediterranean, Int. J. Climatol., 31, 1785–1802, https://doi.org/10.1002/joc.2203, 2011.*

*Pepler, A. S., Dowdy, A. J., van Rensch, P., Rudeva, I., Catto, J. L., and Hope, P.: The contributions of fronts, lows and thunderstorms to southern Australian rainfall, Clim. Dyn., 55, 1489–1505, https://doi.org/10.1007/s00382-020-05338-8, 2020.*

***Reviewer #2***
*This manuscript presents a new approach which combines tracks from individual cyclone tracking algorithms to produce composite / best estimate tracks. Overall the manuscript is interesting and well written and the resultant data sets will be of use to the scientific community.*

We would like to thank the Reviewer #2 for their positive evaluation of the manuscript and for providing insightful comments that greatly improved our work.

*I have three main concerns: (1) how applicable this method is / how necessarily this method is to extra-tropical cyclones in other geographic regions, (2) how physically consistent the resultant composite tracks are and (3) how the benchmark data set was created. These are discussed in more detail below along with some other minor comments:*

***1 Major Comments***
*1. Applicability / relation to extra-tropical cyclones in other geographic regions.*

*(a) The manuscript discusses extra-tropical cyclones and Mediterranean cyclones almost inter-changeably without acknowledging that Mediterranean cyclones are a special subset of extra-tropical cyclones. Better discussion of how Mediterranean cyclones differ from e.g. North Atlantic cyclones should be included in the introduction. This is important because certain characteristics of Mediterranean cyclones make them much harder to track than extra-tropical cyclones in the major storm track regions which raises the question – is this method of composite tracks necessarily in all locations?*

We thank the Reviewer#2 for this comment. In terms of processes, Mediterranean cyclones are not expected to differ from other extratropical low-pressure systems. Cyclogenesis is almost always triggered by baroclinic instability due to an intruding upper tropospheric system such as a trough or a cut-off. The importance of diabatic processes is case-to-case dependent with some cases entirely sustained by diabatic processes while racing to maturity (so called tropical transition). Scales are however different: Mediterranean cyclones are typically of shorter lifetimes, smaller sizes and weaker intensities than cyclones developing over open oceans (Flaounas et al., 2014; Lionello et al., 2016).

Mediterranean cyclones are expected to be harder to track from other extratropical cyclones due to the sharp land-sea transitions and the presence of high mountain chains in the region. Such geographical features create noisy field gradients with several stationary vorticity maxima or pressure minima. Despite this particularity of the Mediterranean region, cyclone tracking methods are hardly consistent in capturing the total number of cyclones in any region of the world. This is rather clear when comparing spatial track densities, number of cyclones per year and similarity scores of different cyclone tracking methods in the framework of the IMILAST project (Neu et al., 2013; their Figs 1&6 and their Table 2). In these regards, composite tracks are potentially very useful in any region of the world. In order to address the comment by the Reviewer the following paragraph in section 6 has been included :

*"In this study we used a distance threshold of 300 km to identify similar cyclone track points. This threshold is indeed adapted to the size of Mediterranean cyclones but it could be also*

*envisaged for cyclones in global applications. For instance, let a rather large mid latitude storm be tracked by several CDTMs in a maritime area over the open oceans. In this case, the cyclonic circulation of the storm could encompass an area with a radius of about 1000 km. Very noisy fields due to high spatial resolutions of input fields would potentially lead the CDTMs to find variable representative centers within such a large cyclonic circulation. However, it would be less likely for all, or most CDTMs, to identify these centers in the edges of the cyclonic circulation, such that they are all 300-500 km far from each other. Therefore, a 300-500 km radius would be reasonable to acquire a composite track, still of high confidence level, that omits the CDTMs that identify cyclone centers in the outer areas of the same system.*

*In the same example, the use of a higher distance threshold (e.g. 1000 km) would produce a composite track point of even higher confidence level that also includes the outlier detected centers. Supposing however that most CDTMs detect cyclone centers relatively close to each other, in both cases of lower and higher distance thresholds, the composite track points would be found in similar average locations. Given the regular splitting and merging of cyclone centers within large cyclonic circulations in the mid-latitudes, a large distance threshold would favor the production of longer composite cyclone tracks and high confidence level. Nevertheless, high distance thresholds would prohibit the early detection of cyclones that start e.g. within frontal areas of large cyclonic circulations. In addition, average track locations would risk to be far from the locations of minima of MSLP, geopotential or maxima of relative vorticity. As a result, our methodological approach is transferable to global or any regional application where a distance threshold of 300 to 500 km would be sufficient to yield meaningful results regardless of the physical characteristics of the tracked systems. Nevertheless, in any case, the quality and robustness of composite tracks depend on the efficient calibration and number of included CDTMs, respectively."*

*(b) The manuscript does not clearly acknowledge that many (although not all) of the tracking algorithms were first developed primarily to identify synoptic-scale cyclones in the major oceanic storm tracks and as such, were not originally designed to be used to identify Mediterranean cyclones. This should be acknowledged and it should be made clearer in the main body of the text which tracking algorithms were specifically designed to identify Mediterranean cyclones.*

In order to better address the Reviewer's comment the following sentence has been added to the section 3.2:

*"It is noteworthy that several CDTMs were developed and substantially tested for studies in the Mediterranean region (Appendix A). This could result in a more favorable calibration for Mediterranean cyclones. Especially concerning the performance of M07, this is the only CDTM that uses relative vorticity as an input field while the manual tracks were identified using MSLP. This inconsistency may also have an impact on the results of Fig. 7."*

*c) Are bogus tracks a much bigger problem in the Mediterranean than in the North Atlantic / Pacific / Southern Hemisphere storm tracks? e.g. is this largely a Mediterranean problem? These are mentioned frequently throughout the manuscript yet no evidence of these are given. Could some examples be included or at least some evidence to show a reader that these really do exist and are a significant problem? In addition, it is my understanding that by*

*default many algorithms tracks many cyclones but then filtering is performed to remove weak / stationary cyclones. Does this pre-existing filtering in many methods remove the bogus tracks?*

We thank the Reviewer for this comment. All CDTMs detect and track cyclones by using specific mathematical and physical criteria. From this point of view, bogus tracks might correspond to shallow or weak atmospheric perturbations but also to segments of intense cyclone tracks in a specific phase of their life cycle. In order to address comments from both Reviewers we added the following sentence in the introduction where a more explicit description of bogus tracks is provided :

*"On the other hand, less strict criteria produce a large number of "bogus tracks", which can be perceived as "false positives", or more generally as CDTM artifacts. Bogus tracks might correspond to persistent weak MSLP perturbations or long-lasting vorticity local maxima due to abrupt wind steering (e.g. close to steep topographic features). More generally they can hardly be interpreted as well-organized vortices."*

Bogus tracks are almost always present in CDTM outputs independently from the region of application. This is reflected by the various number of cyclone tracks in different storm track areas in figures 2 and 3 of Neu et al. (2013; please also refer to previous comment), albeit the definition, distinguishing and comparing bogus tracks from different regions is a rather difficult task. Filtering weak tracked features would be expected to reduce significantly the number of bogus tracks although uncertainty would still remain high -whether filtered tracks correspond to well organized cyclones-. This is rather clear from the different number of cyclone tracks in the panels of Fig. 6 that only consider the deepest extratropical systems of Neu et al. (2013).

Given the geographical complexity of the Mediterranean region and its potential impact on the spatial variability of CDTM input fields, the production of bogus tracks might be more likely to occur in this region (please also refer to comment 1). As an example of bogus tracks, we show in the following figure's panel (a) all tracks from all 10 CDTMs that have at least one track point in the 5-day period of 13-17 September 2020. Panel (a) includes 55 tracks where all methods produce 3 to 11 cyclones within a 5-day period. Panel (b) shows the MSLP evolution of all these 55 cyclones. It is rather clear that several identified cyclones are stationary systems, of irregular and non-smooth tracks with no distinct life stages in panel (b). In some cases deepest track points show values of even more than 1015 hPa.

Several of the systems shown in panels (a) and (b) overlap. These overlapping systems are retained in the composite tracks, shown in panels (c) and (d). Indeed, panels (c) and (d) show much less stationary systems in the Mediterranean basin where confidence levels of more than 7 only retain the track of medicane "Ianos" (September 2020; central Mediterranean), one of the strongest in the region (Lagouvardos et al. 2022).

Lagouvardos, K., Karagiannidis, A., Dafis, S., Kalimeris, A., and Kotroni, V.: Ianos—A Hurricane in the Mediterranean, Bulletin of the American Meteorological Society, 103, E1621–E1636, https://doi.org/10.1175/BAMS-D-20-0274.1, 2022.

[Figure]

Neu, U., Akperov, M. G., Bellenbaum, N., Benestad, R., Blender, R., Caballero, R., Cocozza, A., Dacre, H. F., Feng, Y., Fraedrich, K., Grieger, J., Gulev, S., Hanley, J., Hewson, T., Inatsu, M., Keay, K., Kew, S. F., Kindem, I., Leckebusch, G. C., Liberato, M. L. R., Lionello, P., Mokhov, I. I., Pinto, J. G., Raible, C. C., Reale, M., Rudeva, I., Schuster, M., Simmonds, I., Sinclair, M., Sprenger, M., Tilinina, N. D., Trigo, I. F., Ulbrich, S., Ulbrich, U., Wang, X. L., & Wernli, H. (2013). IMILAST: A Community Effort to Intercompare Extratropical Cyclone Detection and Tracking Algorithms, Bulletin of the American Meteorological Society, 94(4), 529-547.

*2. Physical consistency. A major question I have regarding this method is, is the final produced composite track still physically consistent with the spatial pattern and time evolution of the input data (e.g. with the MSLP data from ERA5)? I would very much like to see some evidence of this in the manuscript e.g. plot the cyclone centre from the composite track and the mean sea level pressure at multiple time steps and see if it looks as physically plausible as one individual track.*

We thank the Reviewer for this comment. As an example of physical consistency of composite tracks and ERA5 data, we revised Fig. 2 of the original submission separating the last two panels and producing a new Fig. 3 with additional information. In the following, we present the new figure and the added text in the manuscript:

*"After building a composite track, all of its composite track points have been assigned to the lower MSLP value within a 2.5 degrees circular area. This operation is adequate to identify different stages of a cyclone's lifecycle (e.g. intensification, mature stage and decay). Figure 3a shows the MSLP evolution of the composite track for the cyclone case in Fig. 2a along with the MSLP evolution of the tracks from the 10 CDTMs. It is rather clear that all tracks*

*reproduce similar dynamical lifecycles suggesting that the composite track can be used for a meaningful analysis of different cyclone stages: from a weak low-pressure system until the decay of deep mature cyclones. For the first cyclone case, Figs 3c and 3e show the MSLP fields for hour times 15 and 35 respectively (vertical lines in Fig. 3a) along with parts of the tracks from all 10 CDTMs that eventually contributed to the production of the composite track in Fig. 2e. The black dot in Figs 3c and 3e shows that the composite track point (black dot) at the times depicted by vertical lines in Fig. 3a is meaningfully close to a local minimum of MSLP. However, this should not be a surprise since all tracks from individual CDTMs are fairly well consistent with each other (Fig. 2a). In the second case, Fig. 3b shows that the composite track, as in Fig. 3a, captures both the deepening and decay stages of the cyclone. For this second case, we select two different times: one at the end of the lifetime of the composite track (time 94 in Fig. 3b) and another one 12 hours later, at time 106. Figure 3d shows that the composite track point is still consistent with a MSLP local minimum, while another minimum is followed by the tracks in the eastern Mediterranean. Figure 3f clearly shows that the cyclone tracks in the eastern Mediterranean are still following the same MSLP local minimum while the composite track ceased to exist along with the local MSLP minimum that it was following in Fig 3d."*

[Figure]

*Figure 3 **a** The MSLP evolution of cyclone tracks for the cyclone case (in coloured lines) of November 2020, shown in Fig. 2a. MSLP of the composite track is overlaid in black color. **c** MSLP field in gray contours at the time of the first vertical line in panel a (1 hP contour intervals where thick contours are drawn every 5 hPa starting from 1000 hPa). Tracks from*

*different CDTMs are shown in coloured lines. Colored dots depict the location of cyclone centers and the black dot depicts the location of the corresponding composite track point. **e** As in **c** but for the time of the second vertical line in panel **a**. **b**, **d**, and **f** as in **a**, **c**, and **e** but for a second cyclone case that took place in November 2018 shown in Fig. 2b.*

*3. Generation of the benchmark data set*
*(a) Line 247. Selection of the 120 cyclones that manual subjective tracks were created for. There is not enough details of how these cyclones were selected nor of these cyclones' properties (life time, intensity) relative to all cyclones. It is stated that these cyclones were selected based on previous case studies. As case studies tend to only be performed on extreme / unusual storms, this suggests that these 120 selected cases are not representative of the whole data set.*

We thank the Reviewer for this comment. Not all cyclones in the list correspond to intense systems but still their number is rather small (117 in 1979-2020) to be considered representative of a 42-year climatology. Nevertheless, the purpose of the subjectively tracked cyclones is to provide insights into the performance of the methods and composite tracks rather than to represent a benchmark dataset. We now include an Appendix B (please see our reply to Reviewer #1) that describes the procedure that we follow to select these 117 cyclones (now provided as supplementary material).

*(b) Line 249 – 255. Information given to meteorologists to identify the tracks manually. I think this approach is somewhat flawed as the meteorologists have only been given the MSLP field which is either the same or less information than the tracking algorithms have. This approach wants the meteorologists to behave like the automated methods, rather than the intelligent meteorologists that they are. In my opinion, the meteorologist should have been allowed many more fields / information to create a true bench-mark data set that really identifies true cyclone tracks (i.e in the way a forecaster would), not only those cyclone tracks which are evident in MSLP.*

We agree with the Reviewer that the production of a reference dataset to assess CDTMs performance would demand thorough and rigorous investigation of weather maps and certainly more than one atmospheric variable. On the other hand, one could argue that this approach would be still flawed since the quality of the produced datasets would be subject to the skill or experience of the meteorologists. In our exercise, we minimize this risk and demonstrate the level of agreement between the subjective tracks since results are "only" dependent on the different perception of cyclone locations given the same atmospheric fields. Behaving like the automated methods, as suggested by the Reviewer, we gain insights into the maximum of agreement that we should expect from the individual CDTMs, i.e. how much agreement with the subjectively tracked cyclones would be deemed acceptable. In order to address the Reviewer's concern we added the following paragraph to Section 2.2:

*"In fact, the subjective production of a reference dataset for the ends of CDTMs assessment would demand thorough and rigorous investigation of weather maps and the use of more atmospheric variables (wind, geopotential height,..). In our more simplistic approach the level of agreement between the subjective tracks is strongly dependent on the different human perceptions of cyclone centers in each expert meteorologist, given the same atmospheric*

*fields. This was done to gain insights into the maximum of agreement that we should also expect from the individual CDTMs, i.e. how much agreement with the subjectively tracked cyclones would be considered as acceptable."*

**2 Minor Comments**
*1. Table 1. Could any additional filtering applied to the tracks (e.g. length or duration of the track) be added to this table. It would better allow a reader to see how the methods differ.*

Following the IMILAST protocol (Neu et al., 2013) each CTDM retains tracks at least 24 hours long (this is stated in section 2). No other post-processing is applied to the tracks.

*2. Line 149, Appendix L618. In the text (line 149) it is written that only cyclone tracks which last for at least 24 hours are retained. When checking the Priestley reference in line 618, it appears that M07 only keeps tracks which last 48 hours (and travel 1000 km). Please resolve this confusion (based on Figure 6 I think tracks that last at least 24 hours are retained). Adding more details to the appendix for M07 would resolve this. It also appears that M07 only keeps tracks if the maximum vorticity exceeds a given threshold. This information should also be added.*

We thank the Reviewer for spotting this error. Indeed we retained only tracks that last at least 24 hours and maximum vorticity must exceed $1\times10^{-5}$ s$^{-1}$. Appendix A has been corrected.

*3. Line 161 – 162. The distance criteria for whether tracks overlap or not is given here as 300 km. This threshold value is justified based on the scale of Mediterranean cyclones. Therefore, would this value need to be different if this method was used for e.g. north Atlantic cyclones which are generally larger but potentially easier to track? Aspects of this new method of combining tracks which are specifically for Mediterranean cyclones need to be more clearly highlighted.*

Please refer to our reply to the Reviewer #2 first major comment.

*4. Line 200 / Figure 2, panels e and f. These panels are hard to read. Would it be possible to only show the composite track / black line and the individual tracks for each tracking algorithm? i.e. the black dots of varying sizes could be removed.*

Figure 2 has been revised accordingly.

*5. Line 307. In my opinion M06 also has a prominent seasonal cycle (albeit with a different phase to M04 and M09) and should be noted here.*

We thank the Reviewer for spotting that. This section has been revised accordingly:

*"... Cyclogenesis tends to be evenly distributed along the year with 7 to 12% per month. Methods M04 and M09 show prominent seasonal cycles with a minimum (maximum) in (summer) spring. On the other hand, several methods (M01, M03, M06 and M10) show a peak of cyclogenesis during the summer months of the year…"*

*6. Line 313 / track density. How was track density computed and is it the same for all 10 methods? Please add these details. Related to this is in Figure 5 the track density goes to the edge of the domain for some methods (e.g. 4, 7, 9) whereas for others (1,3,6,8) there is some halo / gap around the edge. Why is this?*

We thank the Reviewer for spotting this lack in the manuscript. The paragraph has been rephrased as follows:

*"- Cyclone tracks' density: Figure 5 shows different patterns of cyclone track densities, calculated at every grid point as the average count of track points per year within a circular area of radius of 0.5°. It is noteworthy that several CDTMs apply spatial filters to smooth input fields (Appendix A). Therefore, no cyclones are identified at the edges of several panels in Fig. 6 and blank areas depend on the size of the spatial filter. For all CDTMs,..."*

*7. Figure 5. There is some overlap with the longitude labels on the bottom right of each panel. Could this be fixed?*

We thank the Reviewer for spotting the error in the figure. The figure has been redrawn.

*8. Line 333 – 334. This is a confusing and redundant sentence. It is not surprising that the lowest values occur at the time of minimum MSLP. In addition, "mature" stage is not clearly defined and could be misunderstood to mean occluded / after the point of minimum pressure.*

We thank the Reviewer for this comment. In the introduction we added the following sentence:

*"In addition, all produced tracks will tend to be limited to times close to cyclones' mature stage (i.e. time of maximum intensity defined by the track point of lowest MSLP, or highest maximum vorticity) since cyclone centers in early and late stages will be discarded or filtered out by the method's strict criteria and preprocessing procedures."*

And Lines 333-334 have been revised as follows:

*"By design, all CDTMs capture lower MSLP distributions during the cyclone mature stage (middle boxplot in Fig. 6c), defined as the track point of minimum MSLP. However, the overlap of the mature stage distributions of MSLP with those of the initial and decay stages is fairly large. In fact,..."*

*9. Figure 6. To be consistent with the other figures, could method 01 be at the top and M10 at the bottom?*

Figure 6 has been revised accordingly.

*10. Line 508: Suggest replacing "extra-tropical cyclones" with "Mediterranean cyclones" here.*

We thank the Reviewer for the suggestion. As also discussed in the replies above and in the discussions part of the manuscript, the difficulty in producing reference tracks still stands for any extratropical cyclone, including Mediterranean cyclones. We prefer to leave the phrase as it is.

---

## Referee Report (RR1)

Review on "A composite approach to produce reference datasets for extratropical cyclone tracks: Application to Mediterranean cyclones" by Flaounas et al.

Overall, the authors have carefully considered my previous comments and have revised the manuscript in a suitable way. I thank the authors for doing so. I only have a few comments remaining, most very minor. The two exceptions are:

1. In my opinion, my previous comment #1b concerning which tracking algorithms have been designed specifically for Mediterranean cyclones, has not been fully dealt with. This information should be included clearly in the main text, not in an Appendix where it is not particularly clear which methods were designed specifically for Mediterranean cyclones. It could possibly be added to Table 1 as well as mentioned in the text.
2. I still do not think the authors answer my previous comment #1a of whether this method is *necessary* in all regions. The manuscript now discusses whether this method would work and how appropriate it would be and while I appreciate the addition it is not the same thing. However, I also appreciate to answer this thoroughly would require extensive work which is beyond the scope of this manuscript. However, a comment about whether this method is needed / is necessary everywhere could be added to the discussion e.g. near line 710.

**Very minor comments:**

1. Line 101. Could also add coastlines here as well as steep topographic barriers.
2. Line 140. Many types of cyclones elsewhere in the world also cross continental areas so I do not think this is unique to Mediterranean cyclones.
3. Line 153 – 156. Does hourly resolution really make tracking easier? Aren't many tracking algorithms designed to work with 6 hourly data and hence include thresholds based on this?
4. Line 313: "..to gain insights into the maximum of agreement…" There is something missing / not quite right here.
5. Line 334. "...that the complexity of cyclone systems is evolving in time…", I don't agree with this statement (or maybe I misunderstand). I would think it is more likely that decreasing percentages as a function of overlap time is caused by the different methods identifying the start of the cyclones at different times.
6. Line 373 – 378. Is this text really needed? It seems a bit odd here.
7. Line 450. M04 also uses relative vorticity as an input field. This sentence should be more specific to state that M07 is the only method which only uses relative vorticity.
8. Line 463. This is quite repetitive – this has just been stated on line 451.
9. Line 581: "*M01 and M03 contribute to more than half of composite tracks even in datasets with low confidence level*". Do you know what is the reason for this? I find it quite interesting.
10. Line 634. I assume time is in UTC? This could be added here.
11. Line 648. Should cyclone be cyclonic here?
12. Line 692: "in a maritime area over the open oceans". This could be more concise.
13. Line 696. Change "far from" to "away from"
14. Line 785, M06. Need to start a new paragraph here. Something has gone wrong with the formatting here.
15. Line 801. The -5 and -1 here need to be superscripts.
16. Line 828. Title of Appendix B is hard to see, put in bold?

---

## Author Response (AR2)

We would like to thank both Reviewers for their fruitful comments and constructive suggestions. Their contributions have greatly improved this manuscript.

**Reviewer #1**
*The authors have carefully considered the feedback from the previous round of revisions, and the paper is now ready for publication.*

*Technical edits:*
*L306 - This paragraph is very long - perhaps start a new paragraph from "It is noteworthy"*

Agreed. The paragraph has been modified accordingly.

*L620 - The Data Availability section has been misplaced, and should be moved to after the conclusions*

Agreed. The text has been modified accordingly.

*Optional edit: Bogus tracks*

We thank the Reviewer  for this comment and  agree with them that it is not adequate to introduce "Bogus tracks"  in the abstract. The paragraph has been modified to address the comments below.

*Both reviewers questioned the term "bogus tracks", and your responses suggested that many of these may be better understood as weak systems - e.g. Figure 3f shows a clear SLP minimum even after the composite track has ceased. While your wording is ultimately up to you, I recommend you at least remove the term "bogus" from your abstract, saving it for the main text after it has been adequately defined.*
*L40: The same general idea could be conveyed by "More than that, it is typical for CDTMs to produce a non-negligible number of tracks of weak atmospheric features, which do not correspond to large or mesoscale vortices and can differ significantly between CDTMs. Lack of consensus in CDTM outputs and the inclusion of significant amounts of uncertain tracks therein, has long prohibited the production of a commonly accepted reference dataset of extratropical cyclone tracks."*
*L54: The sentence "This suggests…" could be deleted.*
*L60: remove the segment "and including a minimum number of bogus tracks"*

**Reviewer #2**
*Review on "A composite approach to produce reference datasets for extratropical cyclone tracks: Application to Mediterranean cyclones" by Flaounas et al.*

*Overall, the authors have carefully considered my previous comments and have revised the manuscript in a suitable way. I thank the authors for doing so. I only have a few comments remaining, most very minor. The two exceptions are:*

*1. In my opinion, my previous comment #1b concerning which tracking algorithms have been*

*designed specifically for Mediterranean cyclones, has not been fully dealt with. This information should be included clearly in the main text, not in an Appendix where it is not particularly clear which methods were designed specifically for Mediterranean cyclones. It could possibly be added to Table 1 as well as mentioned in the text.*

We agree that it is important to stress this information in the main text. Table 1 has been updated accordingly.

*2. I still do not think the authors answer my previous comment #1a of whether this method is **necessary** in all regions. The manuscript now discusses whether this method would work and how appropriate it would be and while I appreciate the addition it is not the same thing. However, I also appreciate to answer this thoroughly would require extensive work which is beyond the scope of this manuscript. However, a comment about whether this method is needed / is necessary everywhere could be added to the discussion e.g. near line 710.*

Indeed, the discussion we added in the last section about the "300 km distance threshold" does not explicitly address their comment. However, identifying our method as "necessary" -in the Mediterranean or other regions- would be a rather strong statement. To better address the Reviewer's concern, we added the following text at the beginning of the paragraph that discusses the 300 km distance threshold.

*"The conformation of the Mediterranean region triggers peculiar cyclogenesis processes with land-sea contrasts limiting size, intensity and lifetime of perturbations. For those reasons, Mediterranean cyclones are weather systems that bear different characteristics, more than the typical extratropical cyclones developing over the open ocean. However, difficulties in the detection and tracking of extratropical cyclones are also observed in other regions around the globe (Neu et al., 2013). Therefore, we consider the positive aspects of our method to be also applicable in these regions since our approach has no geographical constraints and is not targeting specific cyclone categories. In these regards, the distance threshold of 300 km, that we use to identify similar cyclone track points, is indeed adapted to the size of Mediterranean cyclones but it could be also envisaged for other extratropical cyclones. For instance, let a rather large mid-latitude storm…"*

*Very minor comments:*
*1. Line 101. Could also add coastlines here as well as steep topographic barriers.*

Thank you for the suggestion. We rephrased the sentence as:

*"....(e.g. close to steep topographic barriers and coastlines)."*

*2. Line 140. Many types of cyclones elsewhere in the world also cross continental areas so I do not think this is unique to Mediterranean cyclones.*

Agreed. The sentence has been modified by removing *"when compared to other extratropical cyclones"*.

*3. Line 153 – 156. Does hourly resolution really make tracking easier? Aren't many tracking algorithms designed to work with 6 hourly data and hence include thresholds based on this?*

Agreed. The sentence has been removed .

*4. Line 313: "..to gain insights into the maximum of agreement..." There is something missing / not quite right here.*

Agreed. We removed the following sentence to avoid confusion:

*"This was done to gain insights into the maximum agreement that we should also expect from the individual CDTMs, i.e. how much agreement with the subjectively tracked cyclones would be considered as acceptable."*

*5. Line 334. "...that the complexity of cyclone systems is evolving in time...", I don't agree with this statement (or maybe I misunderstand). I would think it is more likely that decreasing percentages as a function of overlap time is caused by the different methods identifying the start of the cyclones at different times.*

Thank you for this insightful comment. We have revised accordingly:

*"The decreasing percentages as a function of overlap time is caused by how the different methods identify the early stages of the cyclone life cycle."*

*6. Line 373 – 378. Is this text really needed? It seems a bit odd here.*

We agree that this part does not add much to the discussion. We removed these lines.

*7. Line 450. M04 also uses relative vorticity as an input field. This sentence should be more specific to state that M07 is the only method which only uses relative vorticity.*

Agreed and revised accordingly.

*8. Line 463. This is quite repetitive – this has just been stated on line 451.*

Indeed, we removed the phrase in line 463.

*9. Line 581: "M01 and M03 contribute to more than half of composite tracks even in datasets with low confidence level". Do you know what is the reason for this? I find it quite interesting.*

Both methods are among those that identify most cyclones per year (M01 ~300 per year and M03 ~500 per year, Fig5a) and that identify fairly similar areas of high spatial track densities (Fig. 6). These aspects increase the probability of higher similarities among these two CDTMs. Indeed, both M01 and M03 have relatively high similarity scores with respect to other CDTMs in Fig. 9. Since both M01 and M03 do not share the same input meteorological fields, preprocessing and filtering procedures, it is hard to identify the exact reasons for the high similarity scores (that inevitably lead to high participation in composite track datasets). Nevertheless, high similarities might emerge from the fact that 1000 hPa geopotential fields (used by M01) are highly correlated with the ones of MSLP (used by M03) and the fact that

both methods retain only the lowest local minima within a region of 300 (M03) and 500 km (M01).

*10. Line 634. I assume time is in UTC? This could be added here.*

Thank you for the suggestion. The text has been modified accordingly.

*11. Line 648. Should cyclone be cyclonic here?*

Thank you for the correction. The text has been modified accordingly using "cyclonic".

*12. Line 692: "in a maritime area over the open oceans". This could be more concise.*

We removed "maritime area" to make it clearer that we mean a track uninfluenced by continental area:

*"For instance, let a rather large mid-latitude storm be tracked by several CDTMs over the open ocean."*

*13. Line 696. Change "far from" to "away from"*

Done.

*14. Line 785, M06. Need to start a new paragraph here. Something has gone wrong with the formatting here.*

Indeed, this was a typo. Thank you for spotting this.The text has been modified accordingly.

*15. Line 801. The -5 and -1 here need to be superscripts.*

Done.

*16. Line 828. Title of Appendix B is hard to see, put in bold?*

It is now in bold.